# A robust brain network for sustained attention from adolescence to adulthood that predicts later substance use

Yihe Weng[1], Johann Kruschwitz[2,3], Laura M Rueda-Delgado[1], Kathy L Ruddy[1,4], Rory Boyle[1], Luisa Franzen[1,5], Emin Serin[2,6,7], Tochukwu Nweze[8], Jamie Hanson[9], Alannah Smyth[1], Tom Farnan[1], Tobias Banaschewski[10], Arun LW Bokde[11], Sylvane Desrivières[12], Herta Flor[13,14], Antoine Grigis[15], Hugh Garavan[16], Penny A Gowland[17], Andreas Heinz[2], Rüdiger Brühl[18], Jean-Luc Martinot[19], Marie-Laure Paillère Martinot[19,20], Eric Artiges[19,21], Jane McGrath[11], Frauke Nees[10,13,22], Dimitri Papadopoulos Orfanos[15], Tomas Paus[23,24], Luise Poustka[25], Nathalie Holz[10], Juliane Fröhner[26], Michael N Smolka[27], Nilakshi Vaidya[28], Gunter Schumann[27,28], Henrik Walter[2], Robert Whelan[1]*, IMAGEN Consortium

[1]School of Psychology and Global Brain Health Institute, Trinity College Dublin, Dublin, Ireland; [2]Department of Psychiatry and Psychotherapy CCM, Charité – Universitätsmedizin Berlin, corporate member of Freie Universität Berlin, Humboldt-Universität zu Berlin, and Berlin Institute of Health, Berlin, Germany; [3]Collaborative Research Centre (SFB 940) 'Volition and Cognitive Control', Technische Universität Dresden, Dresden, Germany; [4]School of Psychology, Queens University Belfast, Belfast, United Kingdom; [5]Faculty of Psychology and Neuroscience, Maastricht University, Maastricht, Netherlands; [6]Charité – Universitätsmedizin Berlin, Einstein Center for Neurosciences Berlin, Berlin, Germany; [7]Bernstein Center for Computational Neuroscience, Berlin, Germany; [8]Department of Psychology, University of Utah, Salt Lake City, United States; [9]Department of Psychology, Learning Research & Development Center, University of Pittsburgh, Pittsburgh, United States; [10]Department of Child and Adolescent Psychiatry and Psychotherapy, Central Institute of Mental Health, Medical Faculty Mannheim, Heidelberg University, Mannheim, Germany; [11]Discipline of Psychiatry, School of Medicine and Trinity College Institute of Neuroscience, Trinity College Dublin, Dublin, Ireland; [12]Centre for Population Neuroscience and Precision Medicine (PONS), Institute of Psychiatry, Psychology, & Neuroscience, SGDP Centre, King's College London, London, United Kingdom; [13]Institute of Cognitive and Clinical Neuroscience, Central Institute of Mental Health, Mannheim, Heidelberg University, Mannheim, Germany; [14]Department of Psychology, School of Social Sciences, University of Mannheim, Mannheim, Germany; [15]NeuroSpin, CEA, Université Paris-Saclay, Gif-sur-Yvette, France; [16]Departments of Psychiatry and Psychology, University of Vermont, Burlington, United States; [17]Sir Peter Mansfield Imaging Centre School of Physics and Astronomy, University of Nottingham, University Park, Nottingham, United Kingdom; [18]Physikalisch-Technische Bundesanstalt (PTB), Braunschweig and Berlin, Germany; [19]Institut National de la Santé et de la Recherche Médicale, INSERM U 1299 'Trajectoires développementales & psychiatrie', University Paris-Saclay, CNRS; Ecole Normale Supérieure Paris-Saclay, Centre Borelli, Gif-sur-Yvette, France; [20]AP-HP Sorbonne University, Department of Child and Adolescent Psychiatry, Pitié-

*For correspondence:
robert.whelan@tcd.ie

Salpêtrière Hospital, Paris, France; [21]Psychiatry Department, EPS Barthélémy Durand, Etampes, France; [22]Institute of Medical Psychology and Medical Sociology, University Medical Center Schleswig Holstein, Kiel University, Kiel, Germany; [23]Departments of Psychiatry and Neuroscience, Faculty of Medicine and Centre Hosptalier Universitaire Sainte-Justine, University of Montreal, Montreal, Canada; [24]Departments of Psychiatry and Psychology, University of Toronto, Toronto, Canada; [25]Department of Child and Adolescent Psychiatry and Psychotherapy, University Medical Centre Göttingen, Göttingen, Germany; [26]Department of Psychiatry and Neuroimaging Center, Technische Universität Dresden, Dresden, Germany; [27]Centre for Population Neuroscience and Stratified Medicine (PONS), Department of Psychiatry and Neuroscience, Charité Universitätsmedizin Berlin, Berlin, Germany; [28]Centre for Population Neuroscience and Precision Medicine (PONS), Institute for Science and Technology of Brain-inspired Intelligence (ISTBI), Fudan University, Shanghai, China

**Abstract** Substance use, including cigarettes and cannabis, is associated with poorer sustained attention in late adolescence and early adulthood. Previous studies were predominantly cross-sectional or under-powered and could not indicate if impairment in sustained attention was a predictor of substance use or a marker of the inclination to engage in such behavior. This study explored the relationship between sustained attention and substance use across a longitudinal span from ages 14 to 23 in over 1000 participants. Behaviors and brain connectivity associated with diminished sustained attention at age 14 predicted subsequent increases in cannabis and cigarette smoking, establishing sustained attention as a robust biomarker for vulnerability to substance use. Individual differences in network strength relevant to sustained attention were preserved across developmental stages and sustained attention networks generalized to participants in an external dataset. In summary, brain networks of sustained attention are robust, consistent, and able to predict aspects of later substance use.

## eLife assessment

This study presents an **important** finding on the relationship between brain activity related to sustained attention and substance use in adolescence/early adulthood with a large longitudinal dataset. The evidence supporting the claims of the authors is **convincing**. The work will be of interest to cognitive neuroscientists, psychologists, and clinicians working on substance use or addiction.

## Introduction

Sustained attention is a critical cognitive process in daily life, playing a significant role in academic achievement, social communication, and mental health (*Esterman and Rothlein, 2019*) and can be defined as "the focus on performance on a single task over time, with the goal of explaining both the fluctuations within an individual as well as the individual differences in overall ability to maintain stable task performance" (p. 174) (*Esterman and Rothlein, 2019*). Sustained attention notably improves between the ages of 9 and 16 (*Thomson et al., 2022*), concomitant with cognitive maturation and brain development during adolescence (*Paus, 2005*). The functional neuroanatomy of sustained attention involves cingulate, prefrontal, and parietal cortices; supplementary motor area; frontal eye field; and cerebellum (*Bauer et al., 2020*; *Pinar et al., 2018*).

Cross-sectional studies suggest that substance use during adolescence, including cigarette smoking (*Treur et al., 2015*), alcohol consumption (*Ueno et al., 2022*), and cannabis use (*Wallace et al., 2019*), is associated with poorer sustained attention. For instance, adolescents (14–17 years of age) using cannabis a minimum of 4 days per week for at least the last 6 months showed impaired sustained attention in the rapid visual information processing (RVP) task, and in the immediate memory task versus non-users (*Dougherty et al., 2013*). Adolescents (12–17 years of age) in a high

tetrahydrocannabinol (THC, the primary psychoactive component in cannabis) group exhibited lower accuracy on the RVP task than a low THC group (*Shannon et al., 2010*). Cigarette users aged 18–29 showed significant cognitive impairments in sustained attention than non-smokers in the RVP task (*Chamberlain et al., 2012*). A systematic review of the next-day cognitive effects of heavy alcohol consumption demonstrated impairments in sustained attention during alcohol hangovers using meta-analysis (*Yakir et al., 2007*). These findings highlight the negative associations between substance use and sustained attention.

Given the cross-sectional nature of the behavioral and neuroimaging studies above, it remains unclear if impaired sustained attention predates the initiation of substance use and/or if it is a consequence of substance use. Only one longitudinal study (*Harakeh et al., 2012*) has examined the association between sustained attention and cigarette smoking, employing measurements across three waves and involving a large sample of 1797 adolescents. Poor sustained attention, unlike other neurocognitive functions such as working memory, attention flexibility, or perceptual sensitivity, was associated with the increased probability of adolescents subsequently initiating cigarette smoking between ages 11 and 13 and with a higher chance of being a daily smoker by age 16. Harakeh and colleagues' findings suggest that poor sustained attention may precede the onset of cigarette smoking. However, as their study was based on a behavioral level, the neural correlates underlying these associations remain untested.

Although lower sustained attention has been associated with subsequent cigarette smoking, individuals commonly engage in the concurrent use of multiple substances (*Crummy et al., 2020*), perhaps due to shared pathological substrates for substance use. A meta-analysis identified common neural alterations in primary dorsal striatal, and frontal circuits, engaged in reward/salience processing, habit formation, and executive control across various substances (nicotine, cannabis, alcohol, and cocaine) (*Thiele and Bellgrove, 2018*). Those involved in substance use often co-use both cannabis and cigarettes (*Agrawal et al., 2012*; *Hindocha et al., 2016*; *Weinberger et al., 2018*). *Agrawal et al., 2012*, reported that 90% of cannabis users smoke cigarettes during their lifetime, and the widespread co-use of the two may be attributed to genetic sharing (*Agrawal et al., 2010*; *Yadav et al., 2016*) and similar neural mechanisms (*Klugah-Brown et al., 2020*).

Functional brain networks can predict various behavioral traits, such as substance use (*Yip et al., 2019*) and sustained attention (*Rosenberg et al., 2016*). Previous studies (e.g. *Rosenberg et al., 2018*) have used brain connectivity to develop predictive models of sustained attention that can be generalized to healthy and clinical populations. However, while behavioral changes in sustained attention have been documented and functional brain networks that predict substance use have been identified (*Yip et al., 2019*), the underlying change in sustained attention brain networks from adolescence to adulthood and their relation to substance use are relatively less well described. Lower sustained attention has been accompanied by both stronger reductions in neural activity in the visual cortex and stronger recruitment of the right supramarginal gyrus with increasing time on a sustained attention task with central cues in cigarette smokers as opposed to non-smokers (*Vossel et al., 2011*). In a resting-state functional magnetic resonance imaging (fMRI) paradigm, cannabis users aged 16–26 had stronger connectivity between the left posterior cingulate cortex and the cerebellum, correlated with poorer performance on sustained attention/working memory and verbal learning measures (*Ritchay et al., 2021*). Although most brain connectomic research has utilized resting-state fMRI data, functional connectivity (FC) during task performance has demonstrated superiority in predicting individual behaviors and traits, due to its potential to capture more behaviorally relevant information (*Dhamala et al., 2023*; *Greene et al., 2018*; *Yoo et al., 2018*). Specifically, *Zhao et al., 2023*, suggested that task-related FC outperforms both typical task-based and resting-state FC in predicting individual differences. Hence, we applied task-related FC to predict sustained attention over time.

Previous studies found that FC patterns predicted individual differences in sustained attention (*Chen et al., 2022*; *O'Halloran et al., 2018*; *Sripada et al., 2020*), yet relatively little is known about the relationship between brain activity related to sustained attention and substance use over time. A latent change score model can quantify bidirectional longitudinal relations between substance use and both behaviors and brain activity associated with sustained attention, shedding light on how substance use impacts sustained attention and its associated brain activity, and vice versa. In this study, we used task-fMRI from the IMAGEN dataset, a longitudinal study with >1000 participants at each timepoint (ages 14, 19, and 23 years). We first obtained task-related whole-brain connectivity and then

**Table 1.** Demographic information of adolescents in the linear mixed model across three timepoints.

| | Age 14 | Age 19 | Age 23 |
|---|---|---|---|
| N (three timepoints) | 2148 | | |
| Sex (M/F) | 1055/1093 | | |
| Age (years) | 14.4±0.4 | 19±0.7 | 22.6±0.7 |
| Mean FD (mm) | 0.28±0.32 | 0.18±0.17 | 0.18±0.12 |
| GO RT (ms) | 466.6±80 | 400.7±71.8 | 403.9±73.8 |
| ICV | 0.234±0.038 | 0.224±0.051 | 0.217±0.052 |
| Stop RT (ms) | 461.5±114.8 | 360±82.4 | 363.6±78.2 |
| SSD (ms) | 319.3±148.1 | 188.1±132.4 | 190±158.4 |
| SSRT (ms) | 217.8±37.2 | 213.3±43.3 | 216.2±42.6 |
| pOmission (%) | 4.4±10.5 | 2.6±8.6 | 3.7±11.1 |
| pChoiceError (%) | 4.7±6.6 | 4.8±4.7 | 5.2±7.6 |
| pCommission (%) | 47.9±6.3 | 47.5±6 | 47.2±6.9 |

Note: These data pertain to the participants included in the behavioural analyses. N, number of subjects; FD, framewise displacement of MR images; ICV, intra-individual coefficient of variation (assay for sustained attention); SSRT, stop signal reaction time; GO RT, reaction time in Go trials; Stop RT, reaction time in stop fail trials; SSD, stop signal delay; pOmisssion, probability of go omissions (no response); pChoiceError, probability of choice errors on Go trials; pCommission, probability of commission on Stop trials.

used connectome-based predictive modeling (CPM) to predict sustained attention from ages 14 to 23. Additionally, previous cross-sectional and longitudinal studies (*Broyd et al., 2016*; *Harakeh et al., 2012*; *Lisdahl and Price, 2012*) have shown that there are linear relationships between substance use and sustained attention over time. We therefore employed correlation analyses and a latent change score model to estimate the relationship between substance use and both behaviors and brain activity associated with sustained attention. Given the substantial sample size and longitudinal design of Harakeh et al.'s study, we hypothesized that behavioral and predictive networks associated with lower sustained attention would predict increased substance use (particularly cigarette smoking) over time.

## Results

### Behavioral changes over time

Reaction time (RT) variability is a straightforward measure of sustained attention, with increasing variability thought to reflect poor sustained attention. RT variability can be defined as the intra-individual coefficient of variation (ICV), calculated as the standard deviation of Go RT divided by the mean Go RT from Go trials in the stop signal task. Lower ICV indicates better sustained attention. Participants' demographic information for all analyses is shown in *Table 1* (see also *Supplementary file 1a and b*). A linear mixed model analysis showed significant fixed effects of age (i.e. timepoint) on ICV ($F_{1895.3}$ = 51.14, p<0.001) (*Figure 1A*). Post hoc analysis showed that ICV decreased with age: ICV at age 14 was significantly higher than ICV at ages 19 (t=6.535, p<0.001) and 23 (t=10.109, p<0.001). ICV at age 19 was also significantly higher than that at age 23 (t=4.768, p<0.001). The full results of the linear mixed model analysis are shown in *Supplementary file 1c and d*. In addition, we found that individual differences in ICV were significantly correlated between the three timepoints (*Figure 1B* and *Supplementary file 1e*, all p<$2.8e^{-7}$).

### Cross-sectional brain connectivity

This study employed CPM, a data-driven neuroscience approach, to identify three predictive networks – positive, negative, and combined – to predict ICV from brain connectivity. CPM typically uses the strength of the predictive networks to predict individual differences in traits and behaviors. The predictive networks were obtained based on connectivity analyses of the whole brain. Specifically,

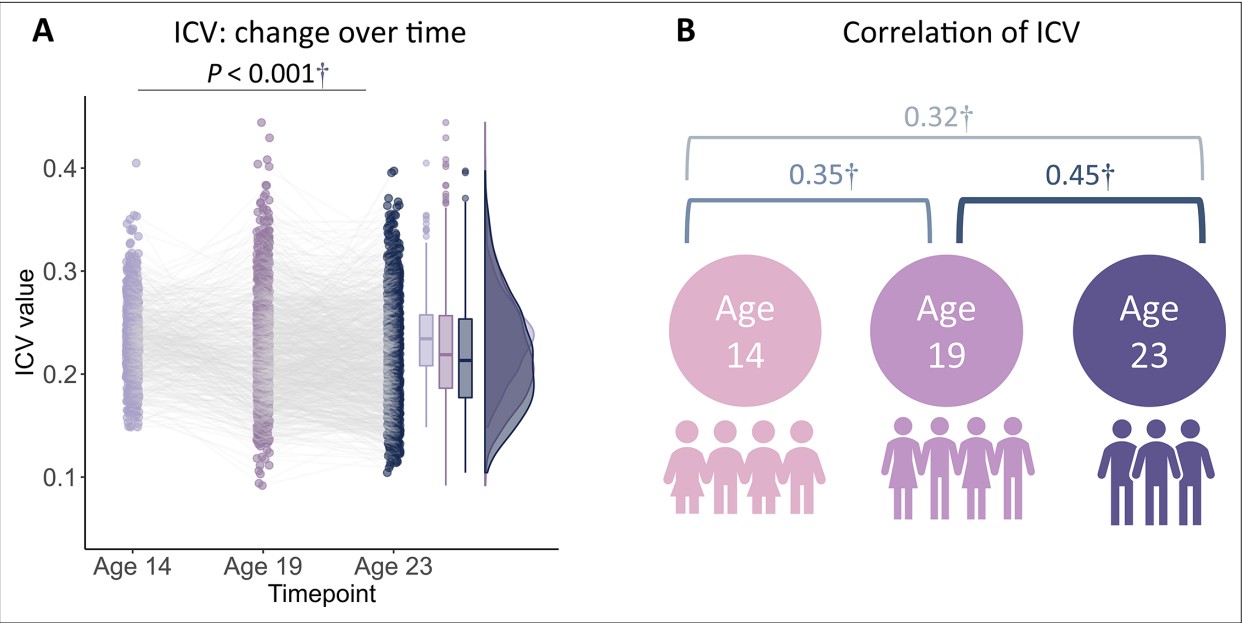

**Figure 1.** Intra-individual coefficient of variation (ICV) changes over time. (**A**) ICV changes over time. (**B**) Correlation of ICV between timepoints within participants. †, p<0.001.

we assessed whether connections between brain areas (i.e. edges) in a task-related FC matrix derived from generalized psychophysiological interaction (gPPI) analysis were positively or negatively correlated with ICV using a significance threshold of p<0.01. These positively or negatively correlated connections were regarded as positive or negative networks, respectively. The network strength of positive networks (or negative networks) was determined for each individual by summing the connection strength of each positively (or negatively) correlated edge. The combined network was determined by subtracting the strength of the negative network from the positive network. We then built a linear model between network strength and ICV in the training set and applied these predictive networks to yield network strength and a linear model in the test set to calculate predicted ICV using k-fold cross-validation (CV).

Positive, negative, and combined networks derived from Go trials significantly predicted ICV: at age 14 (r=0.25, r=0.25, and r=0.28, respectively, all p<0.001) (*Figure 2A*), at age 19 (r=0.27, r=0.25, r=0.28, respectively, all p<0.001) (*Figure 2B*), and at age 23 (r=0.38, r=0.33, and r=0.37, respectively, all p<0.001) (*Figure 2C*). The connectome patterns of predictive networks are shown in *Figure 2D–I*. *Figure 2—figure supplement 1* summarizes the connectivity within and between functional networks and depicts their respective contribution to the predictive network. The above results were validated using 10-fold CV; similar results were obtained when using 5-fold CV and leave-site-out CV (*Supplementary file 1f*). The predictive networks had similar connectome patterns when different exclusion criteria for head motion were used (mean framewise displacement, mean FD <0.2–0.4 mm) (*Figure 3—figure supplements 2–4A*). In addition, we found that network strength of positive, negative, and combined networks derived from Go trials was significantly correlated between the three timepoints (*Supplementary file 1g* , all p<0.003).

Positive, negative, and combined networks derived from Successful stop trials significantly predicted ICV: at age 14 (r=0.22, p<0.001; r=0.12, p=0.017; and r=0.20, p<0.001, respectively) (*Figure 3A*), at age 19 (r=0.19, p<0.001; r=0.15, p=0.001; and r=0.18, p<0.001, respectively) (*Figure 3B*), and at age 23 (r=0.24, r=0.21, and r=0.23, respectively, all p<0.001) (*Figure 3C*). The connectome patterns of predictive networks are shown in *Figure 3D–I*. *Figure 3—figure supplement 1* summarizes the connectivity within and between functional networks and the proportion of brain networks involved in the predictive network. We obtained similar results using a 5-fold CV and leave-site-out CV (*Supplementary file 1e*). The predictive networks had similar connectome patterns when different exclusion criteria for head motion were used (mean FD <0.2–0.4 mm) (*Figure 3—figure supplements 2–4B*). In addition, we found that network strength of positive, negative, and combined networks derived from

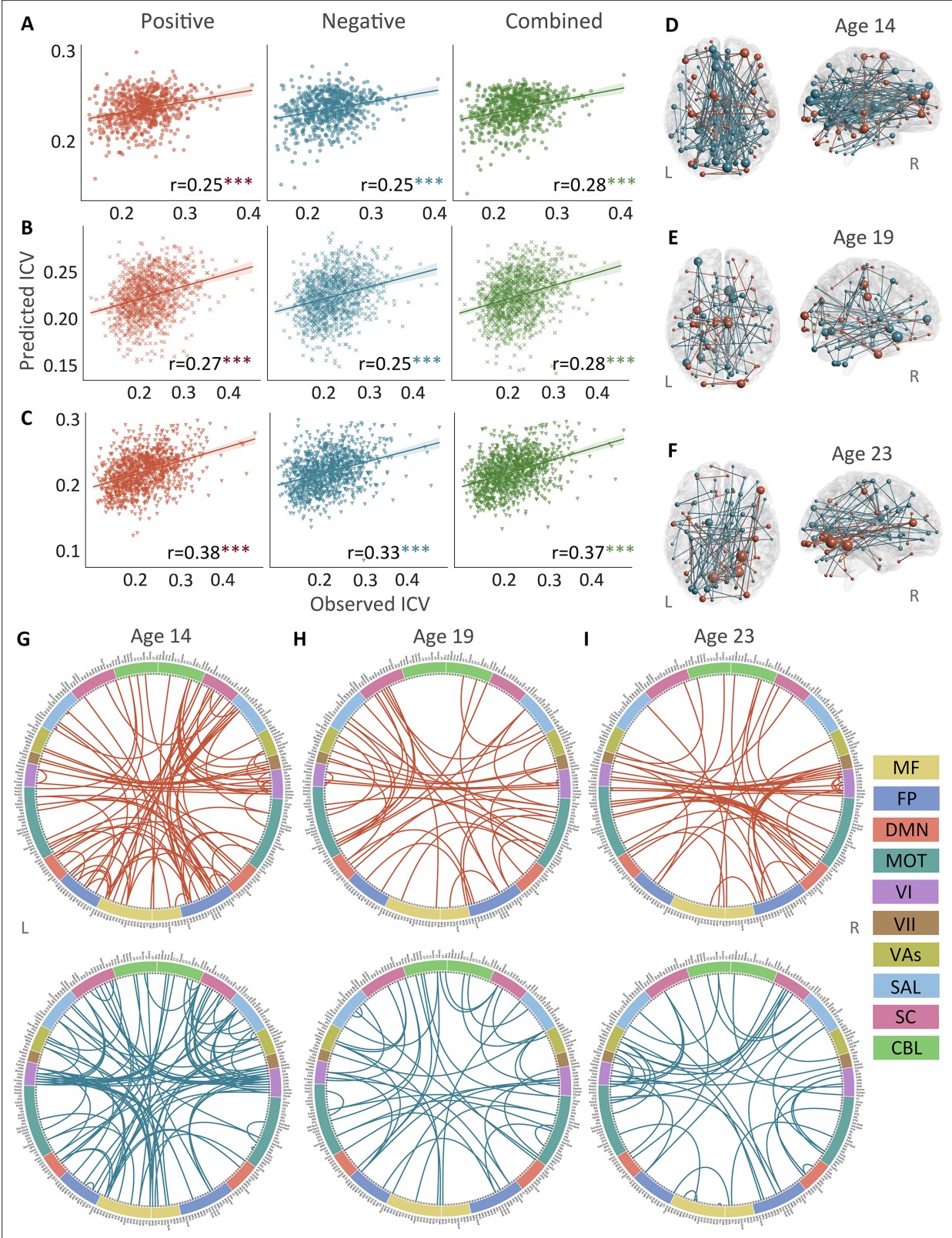

**Figure 2.** The predictive performances and networks of intra-individual coefficient of variation (ICV) per timepoint derived from Go trials. Correlation between observed and predicted ICV in positive, negative, and combined networks at (**A**) age 14, (**B**) age 19, and (**C**) age 23. Predictive networks for ICV are at (**D**) age 14, (**E**) age 19, and (**F**) age 23. Connectome of positive and negative networks of ICV at (**G**) age 14, (**H**) age 19, and (**I**) age 23. The edges depicted above are those selected in at least 95% of cross-validation folds. Red, blue, and green spheres/lines/scatters represent positive, negative, and

*Figure 2 continued on next page*

*Figure 2 continued*

combined networks respectively. MF, medial frontal; FP, frontoparietal; DMN, default mode; MOT, motor; VI, visual I; VII, visual II; VAs, visual association; SAL, salience; SC, subcortical; CBL, cerebellar. R/L, right/left hemisphere. ***, p<0.001.

The online version of this article includes the following figure supplement(s) for figure 2:

**Figure supplement 1.** The predictive networks predicting intra-individual coefficient of variation (ICV) per timepoint derived from Go trials.

Successful stop trials was significantly correlated between the three timepoints (*Supplementary file 1f*, all p<0.001).

To examine the specificity of sustained attention networks identified from CPM analysis, the correlations between the network strength of positive and negative networks and performances from a neuropsychology battery (Cambridge Neuropsychological Test Automated Battery [CANTAB]) (*Fray et al., 1996*) were calculated at each timepoint separately. All positive and negative networks derived from Go and Successful stop trials were significantly correlated with a behavioral assay of sustained attention – the RVP task – at ages 14 and 19 (all p<0.028). Age 23 had no RVP task data in the IMAGEN study. There were sporadic significant correlations between constructs such as delay aversion/impulsivity and negative network strength, for example, but the most robust correlations were with the RVP. Detailed information is shown in Appendix 1 and *Supplementary file 1l*.

### ICV prediction across time

Positive, negative, and combined networks derived from Go trials defined at age 14 predicted ICV at ages 19 (r=0.16, r=0.14, and r=0.16, all p<0.001) (*Figure 4A*, top row) and 23 (r=0.20, r=0.12, and r=0.17, all p<0.001) (*Figure 4A*, middle row). Likewise, positive, negative, and combined networks derived from Go trials defined at age 19 predicted ICV at age 23 (r=0.30, r=0.26, and r=0.31, respectively, all p<0.001) (*Figure 4A*, bottom row).

Positive, negative, and combined networks derived from Successful stop trials defined at age 14 predicted ICV at age 19 (r=0.11, r=0.12, and r=0.13, all p<0.001) (*Figure 4B*, top row) and 23 (r=0.14, r=0.15, and r=0.15, all p<0.001) (*Figure 4B*, middle row). Positive, negative, and combined networks derived from Successful stop trials defined at age 19 predicted ICV at age 23 (r=0.17, r=0.16, and r=0.17, respectively, all p<0.001) (*Figure 4B*, bottom row).

### Generalization of ICV brain networks

We tested if the predictive networks defined at age 23 in IMAGEN would generalize to an external dataset, namely STRATIFY (N = ~300), comprising individuals also aged 23. When applied to the whole STRATIFY sample, positive, negative, and combined networks derived from Go trials at age 23 in IMAGEN predicted ICV in STRATIFY (r=0.34, r=0.34, and r=0.35, respectively, all p<0.001) (*Figure 4C*), as did networks derived from Successful stop trials (r=0.26, r=0.22, and r=0.26, respectively, all p<0.001) (*Figure 4D*).

### Factor analysis of substance use

Exploratory factor analysis on data from the Timeline Followback (TLFB) (*Sobell et al., 1996*), an instrument for measuring the consumption of alcohol, drugs, and smoking for participants, yielded two common factors at age 14 and three common factors at ages 19 and 23. According to the rotated factor loading analysis, at age 14, two common factors were identified, which we labeled as (i) *alcohol* and (ii) *cigarette and cannabis* use (*Cig+CB*). At ages 19 and 23, three common factors were identified, which we labeled as (i) *alcohol*, (ii) *Cig+CB*, and (iii) *drug* (including cocaine, ecstasy, and ketamine) use. Additional details about this data reduction step are shown in *Figure 5—figure supplement 1* and *Supplementary file 1k*.

### Correlation between behavior and brain to cannabis and cigarette use

We calculated the Spearman correlation between ICV/sustained brain activity and TLFB factor score per timepoint and across timepoints. Brain activity was measured by the strength of positive and negative networks predicting sustained attention. The p values were corrected by false discovery rate (FDR) correction (q<0.05). *Figure 5A–C* summarizes the results showing the correlation between ICV/brain activity and Cig+CB per timepoint and across timepoints. *Figure 5A* shows correlations

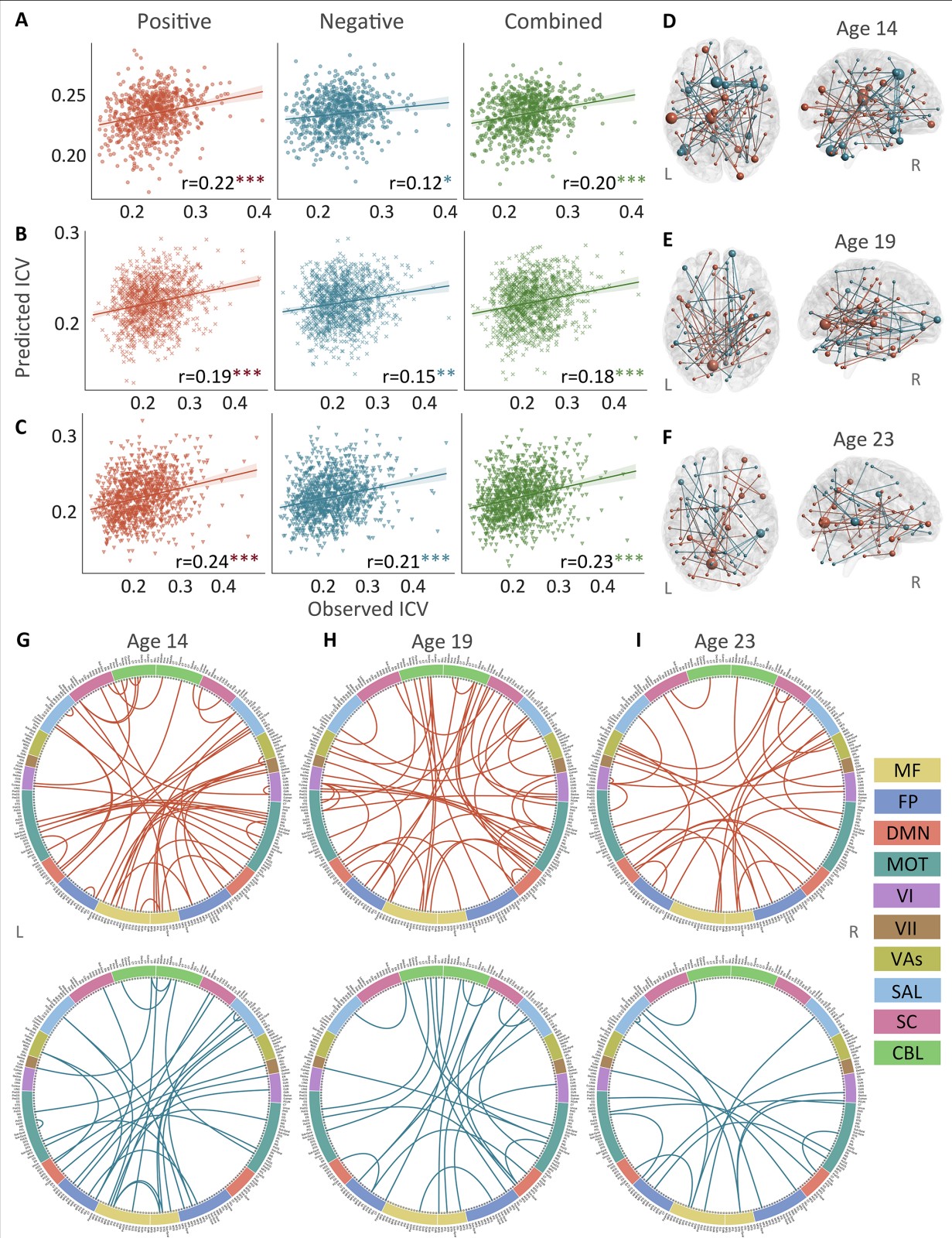

**Figure 3.** The predictive performances and networks of intra-individual coefficient of variation (ICV) per timepoint derived from Successful stop trials. Correlation between observed and predicted ICV in positive, negative, and combined networks at (**A**) age 14, (**B**) age 19, and (**C**) age 23. Predictive networks for ICV are at (**D**) age 14, (**E**) age 19, and (**F**) age 23. Connectome of positive and negative networks of ICV at (**G**) age 14, (**H**) age 19, and (**I**) age 23. The edges depicted above are those selected in at least 95% of cross-validation folds. Red, blue, and green spheres/lines/scatters represent

*Figure 3 continued on next page*

*Figure 3 continued*

positive, negative, and combined networks respectively. MF, medial frontal; FP, frontoparietal; DMN, default mode; MOT, motor; VI, visual I; VII, visual II; VAs, visual association; SAL, salience; SC, subcortical; CBL, cerebellar. R/L, right/left hemisphere. *, p<0.05; **, p<0.01; ***, p<0.001.

The online version of this article includes the following figure supplement(s) for figure 3:

**Figure supplement 1.** The predictive networks predicting intra-individual coefficient of variation (ICV) per timepoint derived from Successful stop trials.

**Figure supplement 2.** Connectome of positive and negative networks predicting intra-individual coefficient of variation (ICV) at age 14 with mean framewise displacement (meanFD) from 0.2 mm to 0.5 mm.

**Figure supplement 3.** Connectome of positive and negative networks predicting intra-individual coefficient of variation (ICV) at age 19 with mean framewise displacement (meanFD) from 0.2 mm to 0.5 mm.

**Figure supplement 4.** Connectome of positive and negative networks predicting intra-individual coefficient of variation (ICV) at age 23 with mean framewise displacement (meanFD) from 0.2 mm to 0.5 mm.

between ICV and Cig+CB (*Supplementary file 1n-o*). ICV was correlated with Cig+CB at ages 19 (Rho = 0.13, p<0.001) and 23 (Rho = 0.17, p<0.001). ICV at ages 14 (Rho = 0.13, p=0.007) and 19 (Rho = 0.13, p=0.0003) were correlated with Cig+CB at age 23. Cig+CB at age 19 was correlated with ICV at age 23 (Rho = 0.13, p=9.38E-05). *Figure 5B* shows correlations between brain activity derived from Go trials and Cig+CB (*Supplementary file 1r-s*). Brain activities of positive and negative networks derived from Go trials were correlated with Cig+CB at age 23 (positive network: $Rho_p$ = 0.12,

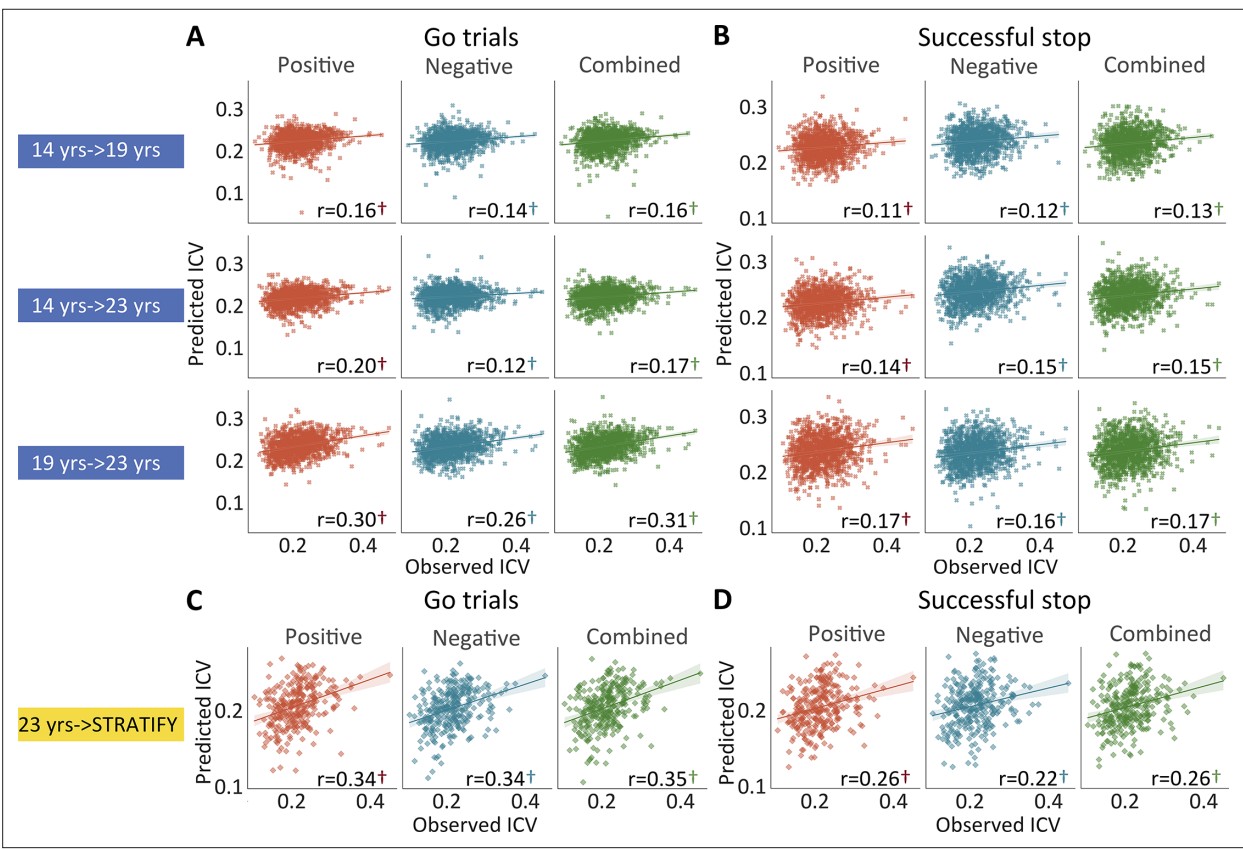

**Figure 4.** The predictive performances of intra-individual coefficient of variation (ICV) across timepoints and generalization in STRATIFY. Predictive performances of ICV (**A**) derived from Go trials and (**B**) derived from Successful stop trials. The top, middle, and bottom rows of (**A**) and (**B**) panels show the predictive performance: using models defined at age 14 to predict age 19 (i.e. 14 years → 19 years), using models defined at age 14 to predict age 23 (i.e. 14 years → 23 years), and using models defined at age 19 to predict age 23 (i.e. 19 years → 23 years) respectively. Generalization of predictive networks predicting ICV defined at age 23 in STRATIFY (i.e. 23 years → STRATIFY) derived from (**C**) Go trials and (**D**) Successful stop trials. The red, blue, and green scatter represent positive, negative, and combined networks. †, p<0.001.

The online version of this article includes the following figure supplement(s) for figure 4:

**Figure supplement 1.** Generalization in subgroups in STRATIFY.

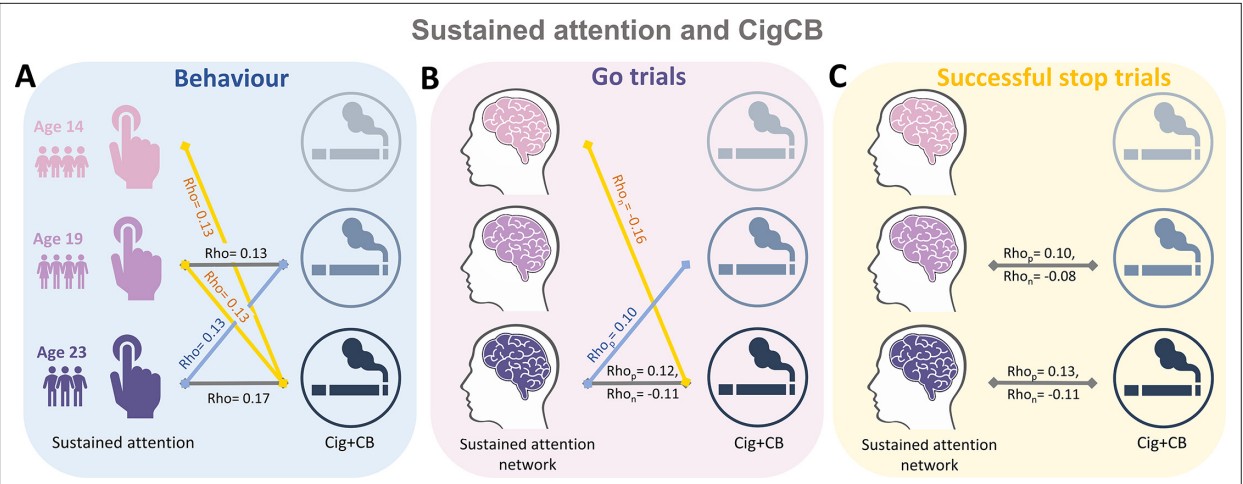

**Figure 5.** Significant correlations between sustained attention and substance use across timepoints (false discovery rate [FDR] correction, q<0.05). (**A**) Correlations between the intra-individual coefficient of variation (ICV) and cigarette and cannabis use (Cig+CB) across timepoints. Correlations between sustained attention network strength and Cig+CB across timepoints (**B**) derived from Go trials and (**C**) derived from Successful stop trials. $Rho_p$: r value between network strength of the positive network. $Rho_n$: r value between network strength of the negative network.

The online version of this article includes the following figure supplement(s) for figure 5:

**Figure supplement 1.** Exploratory factor analysis of Timeline Followback (TLFB) at each timepoint.

**Figure supplement 2.** Significant correlations between sustained attention and substance use across timepoints (false discovery rate [FDR] correction, q<0.05).

p<0.001; negative network: $Rho_n$ = –0.11, p<0.001). Brain activity of the negative network derived from Go trials at age 14 was correlated with Cig+CB at age 23 ($Rho_n$ = –0.16, p=0.001). Cig+CB at age 19 was correlated with brain activity of the positive network derived from Go trials at age 23 ($Rho_p$ = 0.10, p=0.002). *Figure 5C* shows the correlations between brain activity derived from Successful stop and Cig+CB (*Supplementary file 1r-s*). Brain activities of positive and negative networks derived from Successful stop were correlated with Cig+CB at ages 19 (positive network: $Rho_p$ = 0.10, p=0.001; negative network: $Rho_n$ = –0.08, p=0.013) and 23 (positive network: $Rho_p$ = 0.13, p<0.001; negative network: $Rho_n$ = –0.11, p=0.001). No correlation between alcohol use and ICV/brain activity was found after FDR correction. Detailed results on the correlation between ICV/brain activity and substance use can be found in the *Supplementary file 1n-u*.

## Bivariate latent change score model
We used a bivariate latent change score model to explore the relationship between substance use (specifically Cig+CB and alcohol use) and ICV/brain activity. This approach tests for bidirectional associations, examining how substance use at age 14 predicts changes in ICV/brain activity from ages 14 to 23 and vice versa (*Figure 6*). Below, we present the findings regarding the lagged effects of substance use on ICV/brain activity and the lagged effects of ICV/brain activity on substance use (*Table 2*). The p values were corrected by FDR correction (q<0.05).

### Lagged effects of Cig+CB on changes in ICV and brain activity
We examined if Cig+CB use at age 14 predicted the changes in ICV or brain activity (i.e. predictive network strength) associated with sustained attention across ages 14–23. No significance was observed in the lagged effects of Cig+CB on changes in ICV and brain activity (all p>0.172).

### Lagged effects of ICV and brain activity on changes in Cig+CB
We examined if ICV or brain activity associated with sustained attention at age 14 predicted changes in Cig+CB use across ages 14–23. Behaviors and brain activity associated with poor sustained attention predicted a greater increase in subsequent cigarette and cannabis use. Specifically, higher ICV at age 14 predicted a greater increase in Cig+CB from ages 14 to 23 (Std. β=0.12, p<0.001). Higher

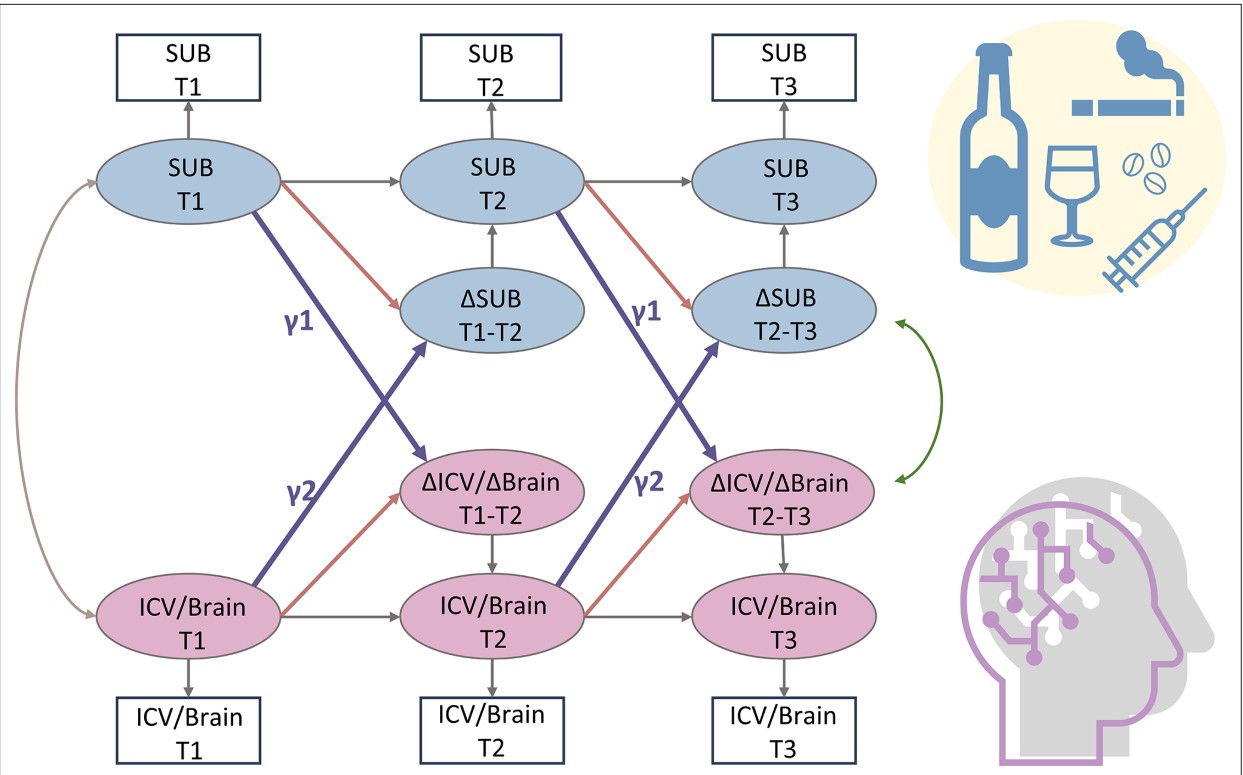

**Figure 6.** A simplified bivariate latent change score model for substance use and ICV/brain activity. SUB, substance use (alcohol, cigarette, and cannabis use); Brain, brain network strength of positive/negative network of sustained attention derived from Go trials/Successful stop trials. ICV, intra-individual coefficient of variation. T1, timepoint 1 (age 14); T2, timepoint 2 (age 19); T3, timepoint 3 (age 23). γ1, lagged effects of substance use on ICV or brain activity. γ2, lagged effects of ICV or brain activity on substance use. The square/circle represents the observation/true score in the model.

sustained attention network strength for positive network derived from Go trials at age 14 predicted a greater increase in Cig+CB from ages 14 to 23 (Std. β=0.09, p=0.006). Lower sustained attention network strength for the negative network, also derived from Go trials at age 14, predicted a greater increase in Cig+CB from ages 14 to 23 (Std. β=–0.09, p=0.006). No other lagged effects of brain activity on changes in Cig+CB remained significant after FDR correction (all p>0.047). *Figure 7* illustrates the changes in raw scores of cigarette and cannabis use from the TLFB for individuals at age 14 with higher sustained attention (i.e. lower ICV, lower strength of positive network, or higher strength of negative network) and lower sustained attention (i.e. higher ICV, higher strength of positive network, or lower strength of negative network).

**Table 2.** Bivariate latent change score model showing the bidirectional association between substance use and ICV/brain networks (false discovery rate corrected).

| | Cig+CB | | Alcohol use | |
|---|---|---|---|---|
| | Lagged effects of Cig+CB (γ1) | Lagged effects of ICV/brain networks (γ2) | Lagged effects of alcohol use (γ1) | Lagged effects of ICV/brain networks (γ2) |
| | Std. β (SE) | Std. β (SE) | Std. β (SE) | Std. β (SE) |
| ICV | 0.017 (0.039) | 0.117 (0.031)*** | 0.005 (0.029) | 0.057 (0.030) |
| SA GT PosNet | –0.026 (0.030) | 0.087 (0.032)** | 0.025 (0.030) | 0.022 (0.036) |
| SA GT NegNet | 0.012 (0.026) | –0.094 (0.035)** | –0.012 (0.030) | –0.059 (0.034) |
| SA SS PosNet | 0.005 (0.025) | 0.070 (0.036) | 0.101 (0.040) | 0.046 (0.039) |
| SA SS NegNet | 0.038 (0.028) | –0.061 (0.031) | –0.003 (0.035) | –0.069 (0.031) |

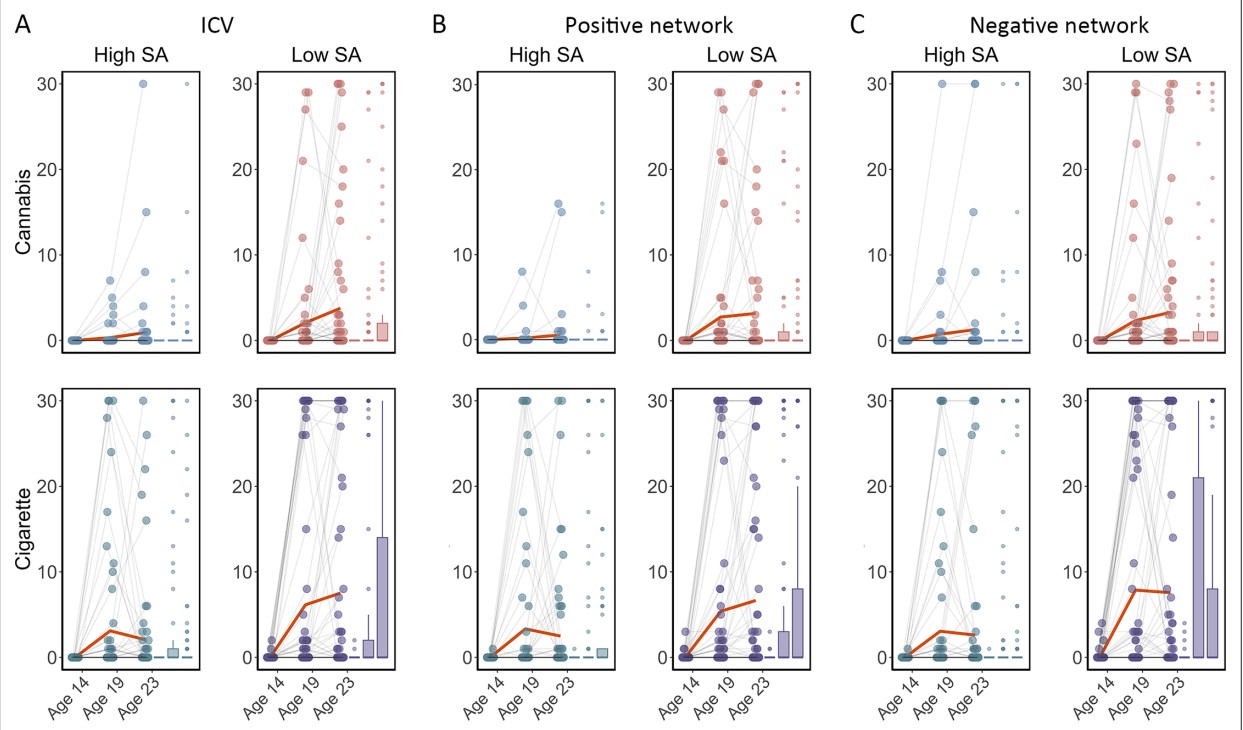

**Figure 7.** Cigarette and cannabis score in Timeline Followback changes in individuals with high sustained attention (High SA) and low sustained attention (Low SA) from ages 14 to 23. Participants were categorized into five equal groups based on the intra-individual coefficient of variation (ICV), strength of positive network, and strength of negative network at age 14. (**A**) Top ICV (Low SA) and bottom ICV (High SA) groups. (**B**) The top strength of the positive network (Low SA) and bottom strength of the positive network (High SA) groups derived from Go trials. (**C**) The top strength of the negative network (High SA) and bottom strength of the negative network (Low SA) groups derived from Go trials. Note that the higher strength of the negative network reflects lower ICV and higher sustained attention.

The online version of this article includes the following figure supplement(s) for figure 7:

**Figure supplement 1.** Cigarette and cannabis score in Timeline Followback change in individuals with good working memory (Good WM) and poor working memory (Poor WM) from ages 14 to 23.

## Association between alcohol use and ICV/brain activity

We examined if alcohol use at age 14 predicted changes in ICV or brain activity associated with sustained attention across ages 14–23, or vice versa. No significant results were found for the lagged effects of alcohol use on changes in ICV and brain activity, nor the lagged effects of ICV and brain activity on changes in alcohol use. The p values were insignificant after FDR correction (all $p > 0.011$).

## Discussion

It is well known that increased substance use, including cigarettes and cannabis, is associated with poorer sustained attention in late adolescence and early adulthood (*Chamberlain et al., 2012*; *Dougherty et al., 2013*). However, previous studies, which were predominantly cross-sectional or under-powered, left a critical question unanswered. That is, was the impairment in sustained attention a predictor of substance use or a marker of the inclination to engage in such behavior? Using a substantial sample size, our results indicate that behavior and brain connectivity associated with poorer sustained attention at age 14 predicted a larger increase in cannabis and cigarette smoking from ages 14 to 23. Furthermore, our findings highlight the robustness of the brain network associated with sustained attention over time, making the latter a potentially useful biomarker for vulnerability to substance use.

## Substance use and the sustained attention network

Our study applied a latent change score model on a large longitudinal dataset, testing the precedence between substance use and sustained attention. In contrast to prior research suggesting that substance use impaired sustained attention (*Broyd et al., 2016*; *Figueiredo et al., 2020*), our results indicate that lower sustained attention also predates substance use. A link between substance use and sustained attention is plausible, given the underlying neurobiology of this sustained attention. Substantial evidence from neuropharmacological studies in rats and humans has shown the modulatory role of neurotransmitters in sustained attention (*Bloomfield et al., 2016*; *Granon et al., 2000*; *Marshall et al., 2019*). Elevated dopamine and noradrenaline levels in the prefrontal cortex lead to improved sustained attention in a dose-dependent manner (*Marshall et al., 2019*). In humans, methylphenidate, a psychostimulant commonly used to treat ADHD, increases both noradrenaline and dopamine signaling and improves sustained attention (*Dockree et al., 2017*). Thus, poorer sustained attention may reflect a lower basal level of dopamine and noradrenaline. More importantly, studies in primates (*Morgan et al., 2002*; *Nader et al., 2006*), rodents (*Dalley et al., 2007*; *Trifilieff et al., 2017*), and humans (*Casey et al., 2014*; *Trifilieff and Martinez, 2014*; *Volkow et al., 2006*) have indicated that low basal dopamine levels are markers of vulnerability for increased drug administration. For example, *Casey et al., 2014*, demonstrated that blunted dopamine release may precede the development of addiction in humans. *Nader et al., 2006*, found a negative correlation between baseline D2 receptor availability and rates of cocaine self-administration in monkeys. Thus, these findings collectively suggest that sustained attention and its brain network could serve as a biomarker of vulnerability to substance use.

These results emphasize the specificity of sustained attention and its associated brain networks, rather than other cognitive abilities, for predicting substance use over time. Unlike sustained attention, no significant differences in cigarette and cannabis use were observed between individuals with lower and higher working memory at baseline during the strategy working memory (SWM) task (*Supplementary file 1w* and *Figure 7—figure supplement 1*). Our results support the behavioral-only findings of a previous study (*Harakeh et al., 2012*), which found that individuals with poorer sustained attention, rather than other cognitive functions, were more likely to initiate smoking cigarettes. Our study goes further by showing that sustained attention brain networks can predict substance use in the future.

## Neural associations between cigarette and cannabis use

We constructed composite scores of substance use. An exploratory factor analysis identified cigarettes and cannabis items as a common factor, aligning with previous studies (*Ferland and Hurd, 2020*; *Hindocha et al., 2016*; *Weinberger et al., 2018*) that indicate concurrent cannabis and cigarette use among users. A national survey in America indicated that 18–23% of cigarette smokers aged 12–17 met the criteria for cannabis use disorder, in contrast to only 2% of non-smoking youth (*Weinberger et al., 2018*). Another national online survey in the UK reported that 80.8% of cigarette smokers engage in cannabis consumption, indicating a prevalent practice of co-administering cannabis and tobacco through smoking (*Hindocha et al., 2021*). Shared genetic factors (*Agrawal et al., 2010*; *Yadav et al., 2016*) and similar neural associations (*Wetherill et al., 2015*) contribute to the co-use of cannabis and cigarettes. *Yadav et al., 2016*, demonstrated a strong and significant genetic correlation between lifetime cannabis use and lifetime cigarette smoking within a large cohort of 32,330 participants, suggesting a high degree of genetic sharing between the two. Using neuroimaging techniques, *Wetherill et al., 2015*, indicated that individuals who used cannabis, smoked tobacco, or engaged in co-use exhibited larger gray matter volumes in the left putamen compared to healthy controls. Both nicotine and cannabis have similar effects on mesolimbic dopaminergic pathways engaged, modulating dopamine release in the striatum (*Bossong et al., 2009*; *Dongelmans et al., 2021*). Collectively, these findings suggest a similar neural association between cigarette and cannabis use.

## Specificity and robustness of sustained attention networks

The brain networks we describe were specific to sustained attention. The strength of the sustained attention brain network was robustly correlated with RVP task performance, a typical sustained attention task, rather than other cognitive measures (*Supplementary file 1l*). Importantly, as highlighted in a previous study (*Cwiek et al., 2022*), emphasizing the importance of generalization in an external dataset, our study found that the sustained attention network derived from Go trials and Successful

stop trials generalized to an external dataset (see further discussion on the generalization in subgroups in STRATIFY in Appendix 1).

We also replicated and extended the developmental pattern of sustained attention and its networks from mid-adolescence to young adulthood. A notable enhancement in sustained attention (i.e. decreased ICV) was observed from ages 14 to 23, as expected (*Fortenbaugh et al., 2015*; *Williams et al., 2005*). Sustained attention networks derived from Go and Successful stop trials predicted behavior at different timepoints, implying that individual differences in sustained attention and associated networks were preserved throughout development. Previously, in neurodiverse youth, attention networks in individuals remained stable across months to years (*Sisk et al., 2022*). *Rosenberg et al., 2020*, also illustrated that the same functional connections predicting overall sustained attention ability also forecasted attentional changes observed over minutes, days, weeks, and months. Here, we contribute to these insights by extending the understanding that attention network stability is not only applicable to neurodiverse populations but also holds in a sizeable cohort of healthy participants. Furthermore, our findings indicate that sustained attention networks remain stable over several years, providing valuable insights into the potential for sustained attention to function as a robust and efficient biomarker for substance use. However, there are still some individual variabilities not captured in this study, which could be attributed to the diversity in genetic, environmental, and developmental factors influencing sustained attention and substance use. Future research should aim to explore these variabilities in greater depth to gain better understanding of the relationship between sustained attention and substance use.

In conclusion, robust sustained attention networks were identifiable from ages 14 to 23. Individual differences in sustained attention network strength were predictable across time. Poorer sustained attention and strength of the associated brain networks at age 14 predicted greater increases in cannabis and cigarette smoking from ages 14 to 23.

## Materials and methods

### Participants

All neuroimaging data and behavioral data were obtained from the IMAGEN study. IMAGEN is a large longitudinal study that recruited over 2000 participants aged 14–23 in Europe (*Kaiser et al., 2022*). This study used the stop signal task fMRI data at ages 14, 19, and 23. In addition, we used an independent dataset STRATIFY as external validation for age 23. STRATIFY (N = ~300) is a sub-dataset within IMAGEN that recruits fMRI data from patients aged 23. Written and informed consent was obtained from all participants by the IMAGEN consortium and the study was approved by the institutional ethics committee of King's College London (PNM/10/11-126), University of Nottingham (D/11/2007), Trinity College Dublin (SPREC092007-01), Technische Universitat Dresden (EK 235092007), Commissariat a l'Energie Atomique et aux Energies Alternatives, INSERM (2007-A00778-45), University Medical Center at the University of Hamburg (M-191/07) and in Germany at medical ethics committee of the University of Heidelberg (2007-024N-MA) in accordance with the Declaration of Helsinki. We followed the exclusion criteria outlined in previous studies (*O'Halloran et al., 2018*; *Whelan et al., 2014*). Participants were excluded from the CPM analysis if they had more than 20% errors on the Go trials (incorrect responses or responses that were too late) or if they had a mean framewise displacement (mean FD)>0.5 mm. Finally, 717 participants at age 14, 1081 participants at age 19, and 1120 participants at age 23 were used to predict ICV. In STRATIFY, 304 participants were used to predict ICV.

### Stop signal task

The stop signal task required participants to respond to a Go signal (arrows pointing left/right) by pressing the left/right button while withholding their response if the Go signal was unpredictably followed by a Stop signal (arrows pointing upward). The Go signal was displayed on the screen for 1000 ms in the Go trials, while the Stop signal appeared for 100–300 ms following the Go signal on average 300 ms later in unpredictable Stop trials. To adjust task difficulty dynamically, we used a tracking algorithm on the delay between the Go signal and Stop signal (stop signal delay, 250–900 ms in 50 ms increments) (*Verbruggen et al., 2019*), to produce 50% successful and 50% unsuccessful inhibition trials. The task at age 14 included 400 Go trials and 80 variable delay Stop trials, with 3 and 7 Go trials between successive Stop trials. The task at ages 19 and 23 consisted of 300 Go trials and 60

variable delay Stop trials. Before the MRI scan, participants also performed a practice session with a block of 60 trials to become familiar with the task. ICV is used to assess sustained attention in this task for each participant. ICV reflects short-term within-person variations in task performance (*O'Halloran et al., 2018*). Specifically, ICV is computed by dividing the standard deviation of Go RT by the mean Go RT. Lower ICV indicates better sustained attention.

## Self-report questionnaires

### Puberty development scale

The puberty development scale (PDS), an 8-item self-report assessment, measures the pubertal development of adolescents (*Petersen et al., 1988*). The PDS evaluates physical development using a 5-point scale where 1 corresponds to prepubertal, 2 to beginning pubertal, 3 to mid-pubertal, 4 to advanced pubertal, and 5 to postpubertal. In addition, the items are adapted for sex, such as voice changes for males or menarche for females.

### Timeline Followback

We used the TLFB, a retrospective self-report instrument that uses a calendar method to evaluate prior substance use consumption over the past 30 days (*Sobell et al., 1996*). The TLFB has strong reliability and validity for assessing alcohol consumption, and we used it to measure the use of alcohol, drugs, and smoking for participants.

## MRI acquisition and pre-processing

Functional MRI data of the stop signal task in the IMAGEN study were collected at eight scan sites (London, Nottingham, Dublin, Mannheim, Dresden, Berlin, Hamburg, and Paris), and data in STRATIFY were collected at three scan sites (Berlin, two scanners in London) with 3T MRI scanners. The MR scanning protocols, cross-site standardization, and quality checks are further described in *Whelan et al., 2012*. All images were obtained using echo-planar imaging (EPI) sequence with the following parameters: repetition time=2.2 s, echo time=30 ms, flip angle = 75°, field of view=224 mm × 224 mm, data matrix = 64 × 64, slice thickness = 2.4 mm with 1 mm slice gap, voxel size = 3.5 mm × 3.5 mm × 4.38 mm, 40 transversal interleaved slices. The MRI data has 444 volumes at age 14 and 320–350 volumes at ages 19 and 23. Standardized hardware was used for visual stimulus presentation (Nordic Neurolab, Bergen, Norway) at all scan sites.

All fMRI data from the IMAGEN study were pre-processed centrally using SPM12 (Statistical Parametric Mapping, http://www.fil.ion.ucl.ac.uk/spm/) with an automated pipeline. The images were corrected for slice timing and then realigned to the first volumes to correct head motions. Participants were excluded from the study if they had a mean FD >0.5 mm. Subsequently, the data were non-linearly transformed to the Montreal Neurological Institute Coordinate System space using a custom EPI template with the voxels resampled at 3 mm× 3 mm ×3 mm resolution. Finally, the images were smoothed with a Gaussian kernel at a full-width-at-half-maximum of 5 mm.

## Generalized psychophysiological interaction analysis

In this study, we adopted gPPI analysis to generate task-related FC matrices and applied CPM analysis to investigate predictive brain networks from adolescents to young adults. PPI analysis describes task-dependent FC between brain regions, traditionally examining connectivity between a seed region of interest (ROI) and the voxels of the whole rest brain. However, this study conducted a gPPI analysis, which is on ROI-to-ROI basis (*Di et al., 2021*), to yield a gPPI matrix across the whole brain instead of just a single seed region. First, we conducted a general linear model (GLM) analysis on the pre-processed fMRI data to examine brain activity during the stop signal task. Two separate GLMs were created for Go trials and Successful stop trials. The Go trials model included three task regressors (Go trials, Failed stop trials, and Successful stop trials) and 36 nuisance regressors, which accounted for factors such as head motion and the signal from white matter and cerebrospinal fluid. The 36 nuisance regressors are 3 translations, 3 rotations, mean white matter signal, mean cerebrospinal fluid signal, mean gray matter signal, their derivatives, and the squares of all these variables. Given the high frequency of Go trials in SST, it is common to treat Go trials as an implicit baseline, as in previous IMAGEN studies (*D'Alberto et al., 2018*; *Whelan et al., 2012*). Hence, we built a separate GLM for Successful stop trials, which included two task regressors (Failed and Successful stop trials) and 36

nuisance regressors. All task regressors were modeled by convolving with the canonical hemodynamic response function (HRF) and high pass filtered (128 s). We then conducted a gPPI analysis across the entire brain using the Shen atlas with 268 regions (*Shen et al., 2013*) for both Go and Successful stop trials. The gPPI analysis involved deconvolving the time series of each ROI with the HRF, multiplying it by the psychological variables of interest to yield a neural level PPI term, and convolving the resulting PPI term with the HRF to obtain the BOLD level PPI effects (*Di and Biswal, 2019*). Separate GLM models were used to estimate the PPI effect of each ROI for Go trials and Successful stop trials, regressing the eigenvariate of the seed ROI. The GLM of the Go trials included one regressor of another ROI eigenvariate, three regressors of task condition, three regressors of the PPI effects, and one contrast term (*Equation 1*). The GLM of Successful stop trials included one regressor of another ROI eigenvariate, two regressors of task condition, two regressors of the PPI effects, and one contrast term (*Equation 2*), shown as follows:

$$Y = \beta_0 + \beta_1 * X_{physio} + \beta_2 * X_{psycho(SS)} + \beta_3 * X_{psycho(FS)} + \beta_4 * X_{psycho(GO)} + \beta_5 * X_{physio} * X_{psycho(SS)} + \beta_6 * X_{physio} * X_{psycho(FS)} + \beta_7 * X_{physio} * X_{psycho(GO)} + \varepsilon \quad (1)$$

$$Y = \beta_0 + \beta_1 * X_{physio} + \beta_2 * X_{psycho(SS)} + \beta_3 * X_{psycho(FS)} + \beta_4 * X_{physio} * X_{psycho(SS)} + \beta_5 * X_{physio} * X_{psycho(FS)} + \varepsilon \quad (2)$$

Note: SS, Successful stop trials; FS, Failed stop trials; GO, Go trials.

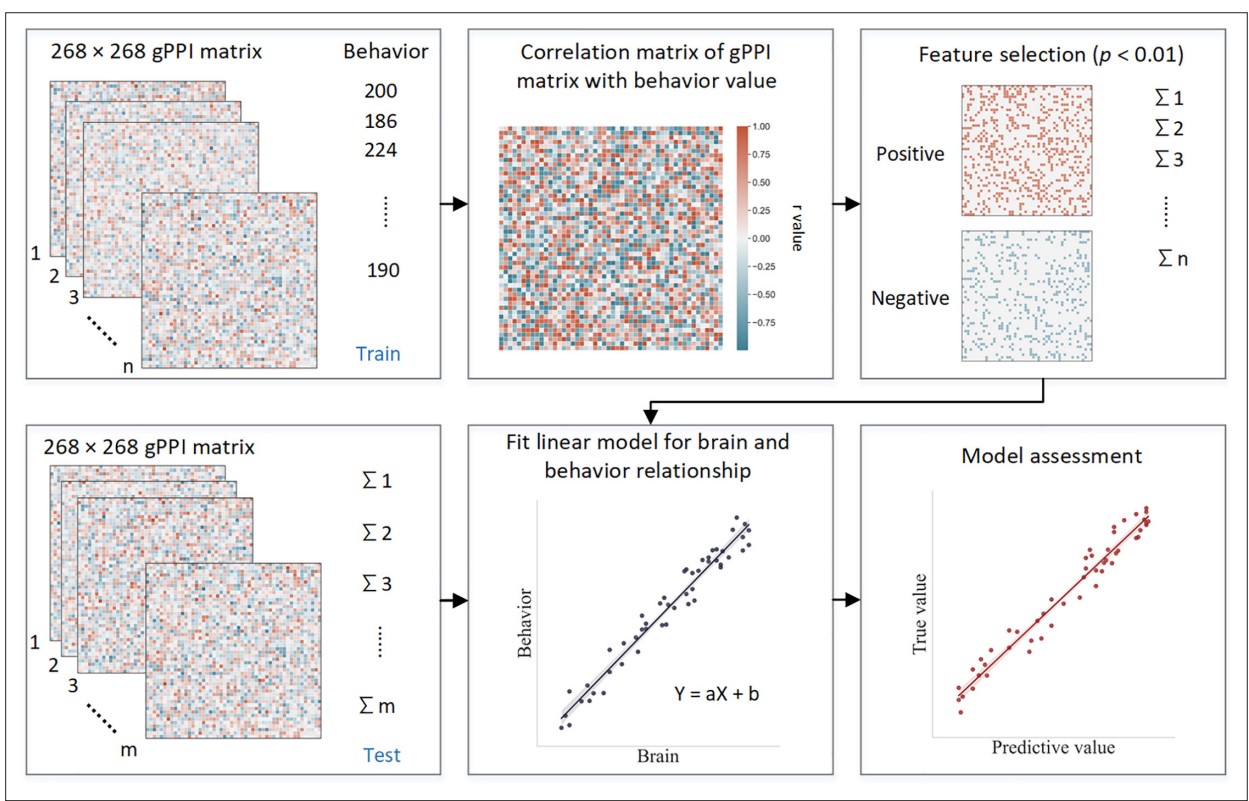

**Figure 8.** Schematic of connectome-based predictive modeling. (**i**) Feature selection. The correlation between each edge in the generalized psychophysiological interaction (gPPI) matrix and the behavioral phenotype is calculated while controlling for several covariates in the training set. These covariates include age, gender, mean framewise displacement (mean FD), scan sites, and mode-centered PDS (only for age 14). The r value with the associated p value for each edge is obtained using partial correlation, and a threshold of p=0.01 is used to select the edges. Positively or negatively correlated edges are regarded as positive or negative networks. Network strength is then calculated by summing the selected edges in the gPPI matrix for both positive and negative networks, as well as by subtracting the strength of the negative network from the strength of the positive network to obtain the combined network strength. (ii) Model building. Linear models are constructed between the network strength of the positive, negative, combined network, and behavioral phenotype in the training set. The network strength is then calculated for each participants in the testing set and input into the predictive model along with covariates to yield a predicted behavioral phenotype (e.g. predicted intra-individual coefficient of variation [ICV]) for each network. (iii) Model validation. The predictive performance is evaluated by calculating the correlation between predicted and observed values.

where Y is the time series of seed ROI, $X_{physio}$ is the time series of another ROI, $X_{psycho}$ is the task design term, and $\varepsilon$ is the residual term. The gPPI analysis was performed across each ROI from the Shen atlas, resulting in a 268*268 gPPI matrix for each participant derived from Go trials and Successful stop trials separately. The matrices were transposed and averaged with the original matrices to yield symmetrical matrices (*Di et al., 2021*), and prepared for further analysis.

## Connectome-based predictive modeling

### ICV prediction

CPM is a data-driven method that can examine individual differences in brain connectivity (*Shen et al., 2017*). CPM identifies pairwise connections between brain regions most highly correlated with a given phenotype. Using the PPI matrix, we employed CPM to predict ICV, for ages 14, 19, and 23. The CPM analysis process includes feature selection, model building, and validation (*Figure 8*). We applied CV to divide all participants into training and testing sets. (i) First, we used partial correlation to calculate the relationship between each edge in the gPPI matrix and behavioral phenotype while controlling several covariates in the training set. These covariates included ages, genders, mode-centered PDS (at age 14 only), mean FD, and scan sites, regarded as a dummy variable. The r value with an associated p value for each edge was obtained, and a threshold p=0.01 (*Feng et al., 2024*; *Ren et al., 2021*; *Yoo et al., 2018*) was set to select edges. The positive or negative correlated edges in feature selection were regarded as positive or negative networks. (ii) Second, we calculated network strength for each participant in the training set by summing the selected edges in the gPPI matrix for both positive and negative networks. We also estimated the network strength of a combined network by subtracting the strength of the negative from the strength of the positive network. (iii) Finally, we constructed predictive models based on the assumption of a linear relationship between network strength of the positive, negative, and combined networks, and behavioral phenotype in the training set. The covariates were also adjusted in this linear model. The network strengths for each participant in the testing set were calculated and input into the predictive model along with the covariates to predict each network's behavioral phenotypes.

### Three CV schemes

We used three CV schemes to test the robustness of predictive performance: k-fold (10-fold and 5-fold) and leave-site-out CV. For the k-fold CV, we randomly divided participants into 10 or 5 approximately equal-sized groups. For each fold, we trained the model on nine or four groups, respectively, and used it to predict the behavioral phenotype of the remaining group. We then assessed the predictive performance by comparing the predicted and observed values. For the leave-site-out CV, we divided participants into eight groups based on their scan site. To account for the random splits of the k-fold CV, we repeated the process 50 times and calculated the average predictive performance for both the 10-fold and 5-fold CV (*Lichenstein et al., 2021*). In addition, we set a 95% threshold for selecting edges present in at least 48 out of 50 iterations to visualize the results. We also ran the CPM analysis with mean FD thresholds of 0.2, 0.3, and 0.4 mm to account for the influence of head motion on the predictive performance. Furthermore, we conducted the CPM analysis using a range of thresholds for feature selection and observed similar results across different thresholds (see Appendix 1, *Supplementary file 1h*). The main text shows the results of the 10-fold CPM. The 5-fold CPM and leave-site-out CV results are shown in Appendix 1.

### Prediction across timepoints and STRATIFY

To assess the ability of models developed at one timepoint to predict ICV at different timepoints, we applied predictive models developed at ages 14 and 19 to predict ICV at subsequent timepoints. Specifically, we used predictive models (including the parameters and selected edges) developed at age 14 to predict ICV at ages 19 and 23. We first calculated the network strength using the gPPI matrix at age 19 or 23 based on the selected edges identified from CPM analysis at age 14. We then used the linear model parameters (slope and intercept) from CPM analysis at age 14 to fit the network strength and predict ICV at age 19 or 23. Finally, we evaluated the predictive performance by calculating the correlation between the predicted and observed values at age 19 or 23. Similarly, we applied models developed at age 19 to predict ICV at age 23. In addition, we examined the generalizability of predictive models at age 23 by applying them to the STRATIFY dataset, which also includes

participants who were 23 years of age. Furthermore, we estimated the predictive performances of ICV across patient groups in the STRATIFY. The correlation between the residual network strength of predictive networks and ICV was calculated across groups in the STRATIFY. The covariates, including age, sex, and mean FD, were regressed for network strength before the correlation analysis. It is worth noting that when applying models developed at one timepoint to predict at another timepoint or to generalize to a different dataset, the model was built using all participants from the timepoint at which the model was developed.

## Statistical analysis

### Exploratory factor analysis

To explore the underlying structure of adolescent substance use, we performed an exploratory factor analysis using principal component extraction (*Gaskin and Happell, 2014*) on TLFB using Predictive Analytics Software (SPSS) version 20. Factor analysis explores the underlying structure of a set of observed variables without imposing a preconceived structure on the outcome. We used six items at age 14 and nine items at ages 19 and 23 of TLFB, including alcohol, tobacco, cannabis, cocaine, ecstasy, and ketamine (as shown in *Supplementary file 1k*). We excluded items assessing the use of other drugs due to high proportions of missing data, standard deviations close to 0, or a Kaiser-Meyer-Olkin (KMO) statistic for individual variables below 0.5, considered the minimum value for a sample to be adequate. The KMO measure of sampling adequacy was 0.66 at age 14, 0.81 at age 19, and 0.77 at age 23. In addition, all Bartlett's tests of sphericity were significant (age 14: $\chi^2(15)=5137.067$, p<0.001; age 19: $\chi^2(36)=5031.641$, p<0.001; age 23: $\chi^2(36)=5106.265$, p<0.001), indicating that there was an underlying correlation structure, and that factor analysis was appropriate. We rotated the factors using the varimax method with kaiser normalization to make it easier to discern the underlying measured constructs.

### Linear mixed model

We constructed a linear mixed model to examine the change in ICV over time using the *lme4* and *lmerTest* packages in RStudio (version: 1.4; http://www.rstudio.com/) and R (version 4.1.1; https://www.r-project.org/). The timepoint was the fixed effect of interest in the model, while the participants was a random effect. Several covariates, including sex, scan sites, mode-center PDS, and age at 14, were also included as fixed effects in the models. The linear mixed model is shown as follows:

$$ICV \sim Timepoint + Sex + Scan\ site + Mode\_center\ PDS + Age\ at\ 14 + (1 \mid Participant) \tag{3}$$

### Correlation between network strength and substance use

To examine the relationship between ICV/brain activity and substance use, we correlated the network strength of predictive networks with the factor scores of substance use at each timepoint and across all three timepoints separately. To control for potential confounders, we calculated residual network strength and residual factor scores by regressing the effects of age, sex, scan sites, mean FD (for network strength), and mode-centered PDS (for age 14). We used Spearman correlation to assess the association between residual network strength and residual TLFB, as their distributions did not follow a normal distribution. We used an FDR correction (q<0.05) for the multiple correlations.

Furthermore, we employed a three-wave bivariate latent change score model using the *lavvan* package in R and RStudio to detect the linear change over time. This model allows us to quantify the longitudinal bidirectional influence between substance use and ICV over time (*Nweze et al., 2023*). Specifically, it facilitated an understanding of whether substance use predicted ICV and its brain activity, and vice versa. The key feature of this model is its ability to assess linear increases or decreases within the same construct across two adjacent waves. Change scores were calculated by regressing the observable score at a given timepoint from the previous timepoint (e.g. ΔCig+CB in T1–T2 or ΔCig+CB in T2–T3, where T1=timepoint 1, T2=timepoint 2, and T3=timepoint 3). Additionally, cross-lagged dynamic coupling (i.e. bidirectionality) was employed to explore individual differences in the relationships between substance use and linear changes in ICV/brain activity, as well as the relationship between ICV/brain activity and linear change in substance use. The model accounted for covariates such as age, sex, and scan sites. For more details about the latent change score model, refer to the reference *Nweze et al., 2023*.

As *Figure 6* shows, the latent change score model was specifically applied to examine the association between substance use and behaviors and brain activity associated with sustained attention. We focused on the relationship between the network strength of positive and negative networks, derived from Go and Successful stop trials, and two types of substance use (Cig+CB and alcohol use). Notably, drug use data were excluded as adolescents at age 14 have no drug score. A total of 10 models were performed, and all model fit indices met the predefined criteria: CFI>0.92, RMSEA<0.05, and SRMR<0.03. An FDR correction (q<0.05) was applied for multiple correlations. It is worth noting that all the correlations between substance use and sustained attention were conducted using the same sample across three timepoints.

## Permutation test

For the CPM analysis, we used a permutation test to assess the significance of the predictive performance, which is the correlation between the observed and predicted values. To generate a null distribution of these correlation values, we randomly shuffled the correspondence of the behavioral data and the PPI matrix of all participants and reran the CPM pipeline with the shuffled data 1000 times. Based on this distribution, we set a threshold of p<0.05 to determine the significance level at 95% for the predictive performance using 10-fold, 5-fold, and leave-site-out CV.

To estimate the significance of the predictive performance across timepoints and the external validation in the STRATIFY dataset, we shuffled the predictive values 1000 times. Then, we correlated the shuffled values with observed values to yield a null distribution of predictive correlation values. We also set a threshold of p<0.05 to determine the significance level at 95% for the predictive performance across timepoints and generalization in STRATIFY.

# Additional information

## Competing interests

Tobias Banaschewski: Served in an advisory or consultancy role for Actelion, Hexal Pharma, Lilly, Lundbeck, Medice, Novartis, Shire. He received conference support or speaker's fee by Lilly, Medice Novartis and Shire. Has been involved in clinical trials conducted by Shire & Viforpharma. Received royalties from Hogrefe, Kohlhammer, CIP Medien, Oxford University Press. The present work is unrelated to the above grants and relationships. Henrik Walter: Received a speaker honorarium from Servier (2014). The other authors declare that no competing interests exist.

## Funding

| Funder | Grant reference number | Author |
| --- | --- | --- |
| China Scholarship Council - Trinity College Dublin Joint Scholarship Programme | 202006750028 | Yihe Weng |
| European Union-funded FP6 Integrated Project IMAGEN | LSHM-CT- 2007-037286 | Gunter Schumann |
| Horizon 2020 | ERC Advanced Grant 'STRATIFY' 695313 | Gunter Schumann |
| Medical Research Council | 'c-VEDA' MR/N000390/1 | Gunter Schumann |
| National Institutes of Health | R01DA049238 | Gunter Schumann |
| Medical Research Foundation and Medical Research Council | MR/R00465X/1 | Gunter Schumann |
| Medical Research Foundation and Medical Research Council | MR/S020306/1 | Gunter Schumann |

| Funder | Grant reference number | Author |
|---|---|---|
| European Union funded project 'environMENTAL' | 101057429 | Gunter Schumann |
| Agence Nationale de la Recherche | ANR-12-SAMA-0004 | Marie-Laure Paillère Martinot<br>Gunter Schumann |
| Science Foundation Ireland | 16/ERCD/3797 | Marie-Laure Paillère Martinot<br>Gunter Schumann<br>Robert Whelan |
| Agence Nationale de la Recherche | AAPG2019 - GeBra | Marie-Laure Paillère Martinot<br>Gunter Schumann |

The funders had no role in study design, data collection and interpretation, or the decision to submit the work for publication.

## Author contributions

Yihe Weng, Conceptualization, Software, Formal analysis, Validation, Investigation, Visualization, Methodology, Writing – original draft, Writing – review and editing; Johann Kruschwitz, Conceptualization, Methodology, Writing – review and editing; Laura M Rueda-Delgado, Kathy L Ruddy, Conceptualization; Rory Boyle, Luisa Franzen, Emin Serin, Methodology; Tochukwu Nweze, Jamie Hanson, Methodology, Writing – review and editing; Alannah Smyth, Tom Farnan, Jane McGrath, Writing – review and editing; Tobias Banaschewski, Arun LW Bokde, Sylvane Desrivières, Herta Flor, Antoine Grigis, Hugh Garavan, Penny A Gowland, Andreas Heinz, Rüdiger Brühl, Jean-Luc Martinot, Marie-Laure Paillère Martinot, Eric Artiges, Frauke Nees, Dimitri Papadopoulos Orfanos, Tomas Paus, Luise Poustka, Nathalie Holz, Juliane Fröhner, Michael N Smolka, Nilakshi Vaidya, Gunter Schumann, Henrik Walter, Investigation; Robert Whelan, Conceptualization, Resources, Data curation, Supervision, Funding acquisition, Validation, Investigation, Methodology, Writing – original draft, Project administration, Writing – review and editing; IMAGEN Consortium, Data curation, Project administration, Resources, Supervision

## Author ORCIDs

Yihe Weng ⓘ https://orcid.org/0009-0005-2628-9492
Laura M Rueda-Delgado ⓘ https://orcid.org/0000-0002-8730-8977
Kathy L Ruddy ⓘ https://orcid.org/0000-0001-5501-0423
Emin Serin ⓘ https://orcid.org/0000-0002-3570-3027
Tom Farnan ⓘ http://orcid.org/0000-0002-7214-4041
Rüdiger Brühl ⓘ https://orcid.org/0000-0003-0111-5996
Jean-Luc Martinot ⓘ https://orcid.org/0000-0002-0136-0388
Dimitri Papadopoulos Orfanos ⓘ https://orcid.org/0000-0002-1242-8990
Michael N Smolka ⓘ https://orcid.org/0000-0001-5398-5569
Robert Whelan ⓘ https://orcid.org/0000-0002-2790-7281

## Ethics

Human subjects: Written and informed consent was obtained from all participants by the IMAGEN consortium and the study was approved by the institutional ethics committee of King's College London (PNM/10/11-126), University of Nottingham (D/11/2007), Trinity College Dublin (SPREC092007-01), Technische Universitat Dresden (EK 235092007), Commissariat a l'Energie Atomique et aux Energies Alternatives, INSERM (2007-A00778-45), University Medical Center at the University of Hamburg (M-191/07) and in Germany at medical ethics committee of the University of Heidelberg (2007-024N-MA) in accordance with the Declaration of Helsinki.

Reviewer #1 (Public review): https://doi.org/10.7554/eLife.97150.3.sa1
Reviewer #2 (Public review): https://doi.org/10.7554/eLife.97150.3.sa2
Reviewer #3 (Public review): https://doi.org/10.7554/eLife.97150.3.sa3
Author response https://doi.org/10.7554/eLife.97150.3.sa4

## Additional files

**Supplementary files**
• Supplementary file 1. Participants' demographic information.

• MDAR checklist

**Data availability**
IMAGEN data are available from a dedicated database: https://imagen2.cea.fr. Due to participant consent restrictions, IMAGEN data cannot be made fully open access. Code for CPM analysis is available at https://osf.io/6ejpd/. Custom code that supports the findings of this study is available at https://github.com/YiheWeng/Weng_eLife_2024_scripts (copy archived at *Weng, 2024*). All data needed to evaluate the conclusions in the paper are present in the paper and/or Appendix 1.

The following previously published dataset was used:

| Author(s) | Year | Dataset title | Dataset URL | Database and Identifier |
|---|---|---|---|---|
| Boyle R, Weng Y, Whelan R | 2023 | 4.4 Studying the connectome at a large scale | https://osf.io/6ejpd/ | Open Science Framework, 6ejpd |

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

# Appendix 1

## Method

### Cambridge Neuropsychological Test Automated Battery

Several CANTAB tasks were used to examine cognitive abilities: the affective go/no-go task (AGN); the RVP task, the SWM task, and the Cambridge guessing task (CGT). Only the participants at age 23 completed the CGT task. The RVP measures sustained attention. The AGN measures inhibitory control in the context of emotionally salient information, the CGT assesses impulsivity, the SWM assesses working memory. Detailed information on CANTAB was described in *Kühn et al., 2020*.

To examine the sustained attention network specificity, we correlated the network strength of predictive networks predicting ICV with CANTAB task at each timepoint separately. To control for potential confounders, we calculated residual network strength and residual performances of three CANTAB tasks by regressing the effects of age, sex, scan sites, mean FD for network strength and mode-centered PDS for age 14. Finally, we used Spearman correlation to assess the association between residual sustained attention network strength and CANTAB performances.

To examine the specificity of sustained attention at baseline in influencing substance use, a two-sample t-test was performed to detect the significant difference in cigarette and cannabis use between high and lower cognition groups at baseline. Participants were categorized into five groups based on the ICV, network strength of positive and negative networks at age 14. The top of participants with the highest ICV/network strength of positive network, or lowest network strength of negative strength comprised the low sustained attention group, while the bottom of participants with lowest ICV/network strength of positive network, or highest network strength of negative strength constituted the high sustained attention group. Cig+CB were then compared between the higher and lower sustained attention groups at each timepoint. We found the significant coupling effect between Cig+CB and network strength derived from Go trials, instead of Successful stop trials. Here, we only tested the difference in Cig+CB using positive and negative network derived from Go trials. We performed the similar analysis by stratifying the participants into higher strategy working memory between error (SWM_BE) group and lower SWM_BE group according to the between error value from the SWM task.

### Generalization in subgroups from STRATIFY

We tested if the predictive networks defined at age 23 in IMAGEN would generalize to distinct patient groups in STRATIFY. STRATIFY includes several subgroups of individuals aged 23 with alcohol use disorder (AUD), major depression disorder (MDD), bulimia nervosa (BN), anorexia nervosa (AN), and 19 healthy controls.

### Dice coefficient

We calculated the Dice coefficient (DC) to quantify the similarity of predictive networks across the three timepoints. A permutation test was also performed to estimate the predictive network similarity's significance. First, we shuffled the ICV at each timepoint and performed feature selection based on random behavioral phenotypes to yield random predictive networks, including positive and negative. Then we calculated DC from predictive networks between each pair of timepoint. These steps were iterated 1000 times to generate a null distribution of DC values. Finally, we set a threshold of $p<0.05$ to determine the significance level at 95% for the similarity of the predictive networks between each timepoint.

### Comparison of predictive networks identified at one timepoint versus another

Steiger's Z value was employed to compare predictive performances of networks identified at different timepoints. This analysis involved comparing the R values derived from networks defined at distinct ages to predict ICV at the same age. For example, we compared the r values of brain networks defined at age 14 when predicting ICV at 19 (i.e. positive network: r=0.25, negative network: r=0.25, combined network: r=0.28) with those R values of brain networks defined at age 19 itself (i.e. positive network: r=0.16, negative network: r=0.14, combined network: r=0.16) derived from Go trials using Steiger's Z test (age 14 → age 19 vs. age 19 → 19). Similarly, comparisons were made between networks defined at age 14 predicting ICV at 23 and those at age 23 predicting

ICV at age 23 (age 14 → age 23 vs. age 23 → 23), as well as between networks defined at age 19 predicting ICV at age 23 and those at age 23 predicting ICV at age 23 (age 19 → age 23 vs. age 23 → age 23). These comparisons were performed separately for Go trials and Successful stop trials.

## CPM analysis using Failed stop trials
We performed another CPM analysis using Failed stop trials using gPPI matrix obtained from the second GLM, described in the main text. The CPM analysis was conducted using 10-fold CV, 5-fold CV, and leave-site-out CV.

## Prediction across timepoints controlling for ICV at age 14
To examine whether connectivity predictors shared variations of sustained attention across timepoints, we applied predictive models developed at ages 14 and 19 to predict ICV at subsequent timepoints controlling for ICV at age 14. Specifically, we used predictive models (including parameters and selected edges) developed at age 14 to predict ICV at ages 19 and 23 separately. First, we calculated the network strength using the gPPI matrix at ages 19 and 23 based on the selected edges identified from CPM analysis at age 14. We then estimated the predicted ICV at ages 19 and 23 by applying the linear model parameters (slope and intercept) obtained from CPM analysis at age 14 to the network strength. Finally, we evaluated the predictive performance by calculating the partial correlation between the predicted and observed values at ages 19 and 23, controlling for ICV at age 14. Similarly, we applied models developed at age 19 to predict ICV at age 23, also controlling for ICV at age 14. To assess the significance of the predictive performance, we used a permutation test, shuffling the predicted ICV values and calculating partial correlation to a general random distribution over 1000 iterations.

## Results
### Specificity of sustained attention network
Sustained attention network strength derived from Go trials and Successful stop trials was significantly correlated with the accuracy of the RVP task (all $p<0.05$) for both negative and positive networks at ages 14 and 19 but not with AGN task performance (all $p>0.05$) (*Supplementary file 1l*). These results suggest that the networks derived from Go trails and Successful stop trials are specific to sustained attention.

No significant difference in Cig+CB was found between high and low sustained attention groups (obtained from both behavior level and brain activity) at age 14 (all $p>0.462$). Higher Cig+CB use was found in low sustained attention group compared to high sustained attention group at age 19 (all $p<0.021$) and age 23 (all $p<0.007$) (*Supplementary file 1v*). In addition, no significant difference in Cig+CB was found at ages 14 ($t=0.11$, $p=0.912$), 19 ($t=1.65$, $p=0.10$), and 23 ($t=1.43$, $p=0.154$) between higher and lower SWM_BE groups (*Supplementary file 1w*).

### CPM predictive performance derived from Failed stop trials
Positive, negative, and combined networks derived from Failed stop trials significantly predicted ICV: at age 14 ($r=0.10$, $p=0.033$; $r=0.19$, $p<0.001$; and $r=0.17$, $p<0.001$, respectively), at age 19 ($r=0.21$, $r=0.18$, and $r=0.21$, all $p<0.001$, respectively), and at age 23 ($r=0.33$, $r=0.35$, and $r=0.36$, respectively, all $p<0.001$). We obtained similar results using a 5-fold CV and leave-site-out CV (*Supplementary file 1f*).

### Predictive network similarity
With respect to Go trials, the mean DC values for the positive and negative networks across all three timepoints were 0.06 and 0.03, respectively, for ICV. With respect to Successful stop trials, the mean DC values for positive and negative networks predicting ICV across all three timepoints were 0.01 and 0.01.

Positive and negative networks predicting ICV derived from Go trials were significantly similar between ages 14 and 19 (DC = 0.03, $p=0.001$ and DC = 0.03, $p<0.001$), and between ages 19 and 23 (DC = 0.06 and DC = 0.04, respectively, all $p<0.001$) (FDR correction, 0.05). The positive network predicting ICV derived from Go trials was significantly similar between ages 14 and 23 (DC = 0.07, $p<0.001$) (FDR correction, 0.05). The mean DC of the positive and negative networks predicting ICV across three timepoints derived from Go trials are 0.06 and 0.03, respectively.

The negative networks predicting ICV derived from Successful stop trials were significantly similar between ages 14 and 19 (DC = 0.03, p=0.001) (FDR correction, q<0.05). The mean DC of the positive and negative networks predicting ICV derived from Successful stop trials was 0.01 and 0.01. Detailed results about DC between each pair of timepoints can be found in **Supplementary file 1m**.

## Generalization in subgroups in STRATIFY

We examined generalization to separate patient cohorts in STRATIFY. Brain networks predicting ICV derived from Go trials defined at age 23 generalized to almost all patient cohorts, including AUD, MDD, BN, and AN (all p<0.05). The prediction for the healthy controls was moderately accurate (r~0.4), although this was not statistically significant due to the small sample size (n=19) for Go trials (**Figure 4—figure supplement 1A**, left panel). However, brain networks predicting ICV derived from Successful stop trials failed to predict ICV in individuals with AUD (p>0.05), although they generalized to other patient groups (**Figure 4—figure supplement 1A**, right panel). Furthermore, the correlations between sustained attention network strength of positive, negative, and combined networks derived from Successful stop trials and ICV in the groups with AUD were in the opposite direction compared with all other groups (**Figure 4—figure supplement 1B**).

## Comparison of predictive performance at different timepoints

Steiger's Z value was used to test if the difference in R values obtained using predictive networks at one timepoint versus another. For positive, negative, and combined networks predicting ICV derived from Go trials at age 19, the R values were higher when using predictive networks defined at 19 than those defined at 14 (Z=3.79, Z=3.39, Z=3.99, all p<0.00071). Similarly, the R values for positive, negative, and combined networks predicting ICV derived from Go trials at age 23 were higher when using predictive networks defined at age 23 compared to those defined at ages 14 (Z=6.00, Z=5.96, Z=6.67, all p<3.47e$^{-9}$) or 19 (Z=2.80, Z=2.36, Z=2.57, all p<0.005).

At age 19, the R value for the positive network predicting ICV derived from Successful stop trials was higher when using predictive networks defined at 19 compared to those defined at 14 (Z=1.54, p=0.022), while the negative and combined networks did not show a significant difference (Z=0.85, p=0.398; Z=2.29, p=0.123). At age 23, R values for the positive and combined networks predicting ICV derived from Successful stop trials were higher when using predictive networks defined at 23 compared to those defined at 14 (Z=3.00, Z=2.48, all p<3.47e$^{-9}$) or 19 (Z=2.52, Z=1.99, all p<0.005). However, the R value for the negative network at age 23 did not significantly differ when using predictive networks defined at 14 (Z=1.80, p=0.072) or 19 (Z=1.48, p=0.138).

## Correlation between drug use and behavior and brain activity

ICV negatively correlated with drug use at age 19 (Rho = −0.11, p=0.001) (**Supplementary file 1n**). ICV at age 23 negatively correlated with drug use at age 19 (Rho = −0.08, p=0.014) (**Supplementary file 1q**). Sustained attention network strength derived from Successful stop trials significantly correlated with drug use at age 19 for the positive network (Rho = −0.12, p<0.001; FDR correction, 0.05) (**Supplementary file 1r**). Sustained attention network strength derived from Successful stop trials at age 23 correlated with drug use at age 19 (positive network: Rho = −0.09, p=0.005; negative network: Rho = 0.09, p=0.007) (**Supplementary file 1u**).

## Predictions across timepoints controlling for ICV at age 14

Positive and combined networks derived from Go trials defined at age 14 predicted ICV at ages 19 (r=0.10, p=0.028; r=0.08, p=0.047) but negative network did not (r=0.06, p=0.119). Positive network derived from Go trials defined at age 14 predicted ICV at age 23 (r=0.11, p=0.013) but negative and combined networks did not (r=0.04, p=0.187; r=0.08, p=0.056). Positive, negative, and combined networks derived from Go trials defined at age 19 predicted ICV at age 23 (r=0.22, r=0.19, and r=0.22, respectively, all p<0.001).

Positive, negative, and combined networks derived from Successful stop trials defined at age 14 predicted ICV at ages 19 (r=0.08, p=0.036; r=0.10, p=0.012; r=0.11, p=0.009) and 23 (r=0.11, p=0.005; r=0.13, p=0.005; r=0.13, p=0.017) respectively. Positive, negative, and combined networks derived from Successful stop trials defined at age 19 predicted ICV at age 23 (r=0.18, r=0.18, and r=0.17, respectively, all p<0.001).

## Discussion

### Sustained attention networks functioning as global activation

Our findings strongly indicate that sustained attention relies on global brain activation (i.e. network strength) rather than specific regions or networks (see also *Zhao et al., 2021*). We observed brain networks associated with high or low sustained attention span in large-scale networks across the cortex, subcortex, and cerebellum across adolescence to adulthood (see *Figure 2—figure supplement 1*, *Figure 3—figure supplement 1*, consistent with *Rosenberg et al., 2020*), instead of being confined to a few key regions. In our study, however, although the edges in the sustained attention networks were significantly similar from ages 14 to 23 (*Supplementary file 1m*), there were relatively few overlapping edges in the predictive networks over time. It is worth noting that DC values depend heavily on the significance threshold applied to the data (*Fröhner et al., 2019*). However, sustained attention network patterns identified could efficiently predict sustained attention for the subsequent timepoints. A prior study (*Cai et al., 2019*) has shown that children aged 9–12 could recruit key nodes (e.g. rIFG and rMFG), eliciting an adult-like global activation pattern that predicted their inhibitory control abilities. Similarly, our findings suggest that adolescents likely exhibit adult-like global activation patterns predicting sustained attention.

### Aberrant sustained attention network in AUD

A notable exception was the failure of the network derived from Successful stop trials to generalize to patients with AUD, requiring higher attention levels. We speculate that activity in the sustained attention network of individuals with AUD might be similar to healthy adults during low cognitive demands but abnormal compared to healthy adults when faced with higher cognitive demands. Evidence from past literature shows that alcohol misuse is associated with attention deficits and dysfunctional neural mechanisms (*Gunn et al., 2018*; *Li et al., 2021*; *Narayan et al., 2021*; *Spear, 2018*). Furthermore, lower activation of parietal and prefrontal cortices has been observed in abstinent patients with AUD compared with healthy controls during visual attention tasks (*Zehra et al., 2019*), suggesting that differences in the attention network of individuals with AUD might underlie attention deficits in AUD.

Although the sustained attention network derived from Successful stop trials seen in healthy controls failed to generalize to AUD, we observed no significant difference in behavioral measures – ICV – between healthy controls and those with AUD. Our results may reflect compensatory mechanisms in AUD that allow these individuals to complete sustained attention tasks, which is consistent with prior studies (*Tapert et al., 2004*; *Zehra et al., 2019*). Compensation manifests as abnormal brain activity while performing normally on the task (*Chanraud et al., 2013*). *Zehra et al., 2019*, found brain activation differences during attention tasks despite no differences in behavioral performance between the abstinent patients with AUD and healthy controls. Previous studies (*Chanraud et al., 2013*; *Squeglia et al., 2009*) pointed out that individuals with AUD might exhibit subtle neural reorganization or compensation to preserve normal cognitive abilities. *Tapert et al., 2004*, found that heavy and light drinkers had similar behavioral performance on a working memory task. Activation differences were found in the parietal and occipital lobes and the cerebellar, indicating subtle neuronal reorganization may occur in AUD. Similarly, a study found that individuals with AUD maintained standard working memory by recruiting other cerebellar-based functional networks to complete the task (*Chanraud et al., 2013*). Successful completion of working memory tasks requires sustained attention (*Myers et al., 2017*). These studies suggest that the failure of the sustained attention network derived from Successful stop trials to generalize AUD in the current study may be due to compensatory mechanisms employed by individuals with AUD while completing tasks requiring high-level sustained attention.

### Specificity of the prediction of predictive networks

We found that task-related function connectivity derived from Go trials, Successful stop trials, and Failed stop trials successfully predicted sustained attention across three timepoints. However, predictive performances of predictive networks derived from Go trials were higher than those derived from Successful stop trials and Failed stop trials. These results suggest that sustained attention is particularly crucial during Go trials when participants need to respond to the Go signal. In contrast, although Successful Stop and Failed Stop trials also require sustained attention, these tasks primarily involve inhibitory control along with sustained attention.

