## [Editor Report · eLife assessment]

This study presents an **important** finding on the relationship between brain activity related to sustained attention and substance use in adolescence/early adulthood with a large longitudinal dataset. The evidence supporting the claims of the authors is **convincing**. The work will be of interest to cognitive neuroscientists, psychologists, and clinicians working on substance use or addiction.

---

## [Referee Report · Reviewer #1 (Public review)]

This study explored the relationship between sustained attention and substance use from ages 14 to 23 in a large longitudinal dataset. They found behaviour and brain connectivity associated with poorer sustained attention at age 14 predicted subsequent increase in cannabis and cigarette smoking from ages 14-23. They concluded that the brain network of sustained attention is a robust biomarker for vulnerability to substance use. The big strength of the study is a substantial sample size and validation of the generalization to an external dataset. In addition, various methods/models were used to prove the relationship between sustained attention and substance use over time.

---

## [Referee Report · Reviewer #2 (Public review)]

Weng and colleagues investigated the relationship between sustained attention and substance use in a large cohort across three longitudinal visits (ages 14, 19, and 23). They employed a stop signal task to assess sustained attention and utilized the Timeline Followback self-report questionnaire to measure substance use. They assessed the linear relationship between sustained attention-associated functional connections and substance use at an earlier visit (age 14 or 19). Subsequently, they utilized this relationship along with the functional connection profile at a later age (age 19 or 23) to predict substance use at those respective ages. The authors found that connections in association with reduced sustained attention predicted subsequent increases in substance use, a conclusion validated in an external dataset. Altogether, the authors suggest that sustained attention could serve as a robust biomarker for predicting future substance use.

This study by Weng and colleagues focused on an important topic of substance use prediction in adolescence/early adulthood.

---

## [Referee Report · Reviewer #3 (Public review)]

Summary:

Weng and colleagues investigated the association between attention-related connectivity and substance use. They conducted a study with a sizable sample of over 1,000 participants, collecting longitudinal data at ages 14, 19, and 23. Their findings indicate that behaviors and brain connectivity linked to sustained attention at age 14 forecasted subsequent increases in cigarette and cannabis use from ages 14 to 23. However, early substance use did not predict future attention levels or attention-related connectivity strength.

Strengths:

The study's primary strength lies in its large sample size and longitudinal design spanning three time-points. A robust predictive analysis was employed, demonstrating that diminished sustained attention behavior and connectivity strength predict substance use, while early substance use does not forecast future attention-related behavior or connectivity strength.

---

## [Author Response]

The following is the authors’ response to the original reviews.

**Reviewer #1 (Recommendations For The Authors):**
Although the manuscript is well organized and written, it could be largely improved and therefore made more plausible and easier to read. See my point-by-point comments listed below:(1) The introduction section is a bit overloaded with some unnecessary information. For example, the authors discussed the relationship between neurotransmitters in the prefrontal and striatum and substance use/sustained attention. However, the results are related to neither the neurotransmitters nor the striatum. In addition, there is a contradictory description about neurotransmitters there, Nicotine/THC leads to increased neurotransmitters, and decreased neurotransmitters is related to poor sustained attention. Does that mean that the use of Nicotine/THC could increase sustained attention?

Thanks for this insightful question. We understand your concern regarding the seemingly contradictory statements about neurotransmitters and sustained attention. Previous studies have shown that acute administration of nicotine can improve sustained attention (Lawrence et al., 2002; Potter and Newhouse, 2008; Valentine and Sofuoglu, 2018; Young et al., 2004). On the other hand, the acute effects of smoking cannabis on sustained attention are mixed and depend on factors such as dosage and individual differences (Crean et al., 2011). For instance, a previous study (Hart et al., 2001) found that performance on a tracking task, which requires sustained attention, was found to improve significantly after smoking cannabis with a high dose of THC, albeit in experienced cannabis users. However, chronic substance use, including nicotine and cannabis, has been associated with impaired sustained attention (Chamberlain et al., 2012; Dougherty et al., 2013).

To address your concerns and improve clarity and succinctness of the Introduction, we have removed the description of neurotransmitters from the Introduction. This revision should make the introduction more concise and focus on the direct relationships pertinent to our study.

(2) It is a bit hard to follow the story for the readers because the Results section went straight into detail. For example, the authors directly introduced that they used the ICV from the Go trials to index sustained attention without basic knowledge about the task. Why use the ICV of Go trials instead of other trials (i.e., successful stop trials) as an index of sustained attention? I suggest presenting the subjects and task details about the data before the detailed behavioral results. The results section should include enough information to understand the presenting results for the readers, rather than forcing the reader to find the answer in the later Methods section.

We appreciate your suggestion to provide more context about the task and ICV before diving into the detailed behavioural results.

We used the ICV derived from the Go trials instead of Success stop trials as an index of sustained attention, based on the nature of the stop-signal task and the specific data it generates. Previous studies have indicated that reaction time (RT) variability is a straightforward measure of sustained attention, with increasing variability thought to reflect poorer ability to sustain attention (Esterman and Rothlein, 2019). RT variability is defined as ICV, calculated as the standard deviation of mean Go RT divided by the mean Go RT from Go trials (O'Halloran et al., 2018). The stop signal task includes both Go trials and stop trials. During Go trials, participants are required to respond as quickly and accurately as possible to a Go signal, allowing for the recording of RT for calculating ICV. In contrast, stop trials are designed to measure inhibitory control, where successful response inhibition results in no RT or response recorded in the output. Therefore, Go trials are specifically used to assess sustained attention, while Stop trials primarily assess inhibitory control (Verbruggen et al., 2019).

We acknowledge the importance of providing this contextual information within the Results section to enhance reader understanding. We have added this information before presenting the behavioural results on Page 6.

Results

(1) Behavioural changes over time

Reaction time (RT) variability is a straightforward measure of sustained attention, with increasing variability thought to reflect poor sustained attention. RT variability is defined as intra-individual coefficient of variation (ICV), calculated as the standard deviation of mean Go RT divided by the mean Go RT from Go trials in the stop signal task. Lower ICV indicates better sustained attention.

(3) The same problem for section 2 in the Results. What are the predictive networks? Are the predictive networks the same as the networks constructed based on the correlation with ICV? My intuitive feeling is that they are the circular analyses here. The positive/negative/combined networks are calculated based on the correlation between the edges and ICV. Then the author used the network to predict the ICV again. The manipulation from the raw networks (I think they are based on PPI) to the predictive network, and the calculation of the predicted ICV are all missing. The direct exposure of the results to the readers without enough detailed knowledge made everything hard to digest.

We thank the Reviewer for the insightful comment. We agree with the need for more clarity regarding the predictive networks and the CPM analysis before presenting results. CPM, a data-driven neuroscience approach, is applied to predict individual behaviour from brain functional connectivity (Rosenberg et al., 2016; Shen et al., 2017). The CPM analysis used the strength of the predictive network to predict the individual difference in traits and behaviours. CPM includes several steps: feature selection, feature summarization, model building, and assessment of prediction significance (see Fig. S1).

During feature selection, we assessed whether connections between brain areas (i.e., edges) in a task-related functional connectivity matrix (derived from general psychophysiological interaction analysis) were positively or negatively correlated with ICV using a significance threshold of P < 0.01. These positively or negatively correlated connections are regarded as positive or negative network, respectively. The network strength of the positive network (or negative network) was determined in each individual by summing the connection strength of each positively (or negatively) correlated edge. The combined network was determined by subtracting the strength of the negative network from the positive network. Next, CPM built a linear model between the network strength of the predictive network and ICV. This model was initially developed using the training set. The predictive networks were then applied to the test set, where network strength was calculated again, and the linear model was used to predict ICV using k-fold cross-validation. Following your advice, we have updated it in the Results section to include these details on Page 7.

Results

(2) Cross-sectional brain connectivity

This study employed CPM, a data-driven neuroscience approach, to identify three predictive networks— positive, negative, and combined— that predict ICV from brain functional connectivity. CPM typically uses the strength of the predictive networks to predict individual differences in traits and behaviors. The predictive networks were obtained based on connectivity analyses of the whole brain. Specifically, we assessed whether connections between brain areas (i.e., edges) in a task-related functional connectivity matrix derived from generalized psychophysiological interaction analysis were positively or negatively correlated with ICV using a significance threshold of P < 0.01. These positively or negatively correlated connections were regarded as positive or negative network, respectively. The network strength of positive networks (or negative networks) was determined for each individual by summing the connection strength of each positively (or negatively) correlated edge. The combined network was determined by subtracting the strength of the negative network from the positive network. We then built a linear model between network strength and ICV in the training set and applied these predictive networks to yield network strength and a linear model in the test set to calculate predicted ICV using k-fold cross validation.

(4) The authors showed the positive/negative/combined networks from both Go trials and successful stop trials can predict the ICV. I am wondering how the author could validate the specificity of the prediction of these positive/negative/combined networks. For example, how about the networks from the failed stop trials?

We appreciate the opportunity to clarify the specificity of the predictive networks identified in our study. Here is a more detailed explanation of our findings and their implications.

To validate the specificity of the sustained attention network identified from CPM analysis, we calculated correlations between the network strength of positive and negative networks and performances from a neuropsychology battery (CANTAB) at each timepoint separately. CANTAB includes several tasks that measure various cognitive functions, such as sustained attention, inhibitory control, impulsivity, and working memory. We found that all positive and negative networks derived from Go and Successful stop trials significantly correlated with a behavioural assay of sustained attention – the rapid visual information processing (RVP) task – at ages 14 and 19 (all P values < 0.028). Age 23 had no RVP task data in the IMAGEN study. There were sporadic significant correlations between constructs such as delay aversion/impulsivity and negative network strength, for example, but the correlations with the RVP were always significant. This demonstrates that the strength of the sustained attention brain network was specifically and robustly correlated with a typical sustained attention task, rather than other cognitive measures. The results are described in the main text on Page 8 and shown in Supplementary materials (Pages 1 and 3) and Table S12.

In addition, we conducted a CPM analysis to predict ICV using gPPI under Failed stop trials. Our findings showed that positive, negative, and combined networks derived from Failed stop trials significantly predicted ICV: at age 14 (r = 0.10, P = 0.033; r = 0.19, P < 0.001; and r = 0.17, P < 0.001, respectively), at age 19 (r = 0.21; r = 0.18; and r = 0.21, all P < 0.001, respectively), and at age 23 (r = 0.33, r = 0.35, and r = 0.36, respectively, all P < 0.001). Similar results were obtained using a 5-fold CV and leave-site-out CV.

Our analysis further showed that task-related functional connectivity derived from Go trials, Successful Stop trials, and Failed Stop trials could predict sustained attention across three timepoints. However, the predictive performances of networks derived from Go trials were higher than those from Successful Stop and Failed Stop trials. This suggests that sustained attention is particularly crucial during Go trials when participants need to respond to the Go signal. In contrast, although Successful Stop and Failed Stop trials also require sustained attention, these tasks primarily involve inhibitory control along with sustained attention.

Taken together, these findings underscore the specificity of the predictive networks of sustained attention. We have updated these results in the Supplementary Materials (Pages 3-5 and Page 7):

Method

CPM analysis using Failed stop trials

We performed another CPM analysis using Failed stop trials using gPPI matrix obtained from the second GLM, described in the main text. The CPM analysis was conducted using 10-fold CV, 5-fold CV and leave-site-out CV.

Results

CPM predictive performance under Failed stop trials

Positive, negative, and combined networks derived from Failed stop trials significantly predicted ICV: at age 14 (r = 0.10, P = 0.033; r = 0.19, P < 0.001; and r = 0.17, P < 0.001, respectively), at age 19 (r = 0.21; r = 0.18; and r = 0.21, all P < 0.001, respectively), and at age 23 (r = 0.33, r = 0.35, and r = 0.36, respectively, all P < 0.001). We obtained similar results using a 5-fold CV and leave-site-out CV (Table S6).

Discussion

Specificity of the prediction of predictive networks

We found that task-related function connectivity derived from Go trials, Successful stop trials, and Failed stop trials successfully predicted sustained attention across three timepoints. However, predictive performances of predictive networks derived from Go trials were higher than those derived from Successful stop trials and Failed stop trials. These results suggest that sustained attention is particularly crucial during Go trials when participants need to respond to the Go signal. In contrast, although Successful Stop and Failed Stop trials also require sustained attention, these tasks primarily involve inhibitory control along with sustained attention.

(5) The author used PPI to define the connectivity of the network. I am not sure why the author used two GLMs for the PPI analysis separately. In the second GLM, Go trials were treated as an implicit baseline. What does this exactly mean? And the gPPI analysis across the entire brain using the Shen atlas is not clear. Normally, as I understand, the PPI/gPPI is conducted to test the task-modulated connectivity between one seed region and the voxels of the whole rest brain. Did the author perform the PPI for each ROI from Shen atlas? More details about how to use PPI to construct the network are required.

Thank you for your insightful questions. Here, we’d like to clarify how we applied generalized PPI across the whole brain using the Shen atlas and why we used two separate GLMs for the gPPI analysis.

Yes, PPI is conducted to test the task-modulated connectivity between one seed region and other brain areas. This method can be both voxel-based and ROI-based. In our study, we performed ROI-based gPPI analysis using Shen atlas with 268 regions. Specifically, we performed the PPI on each seed region of interest (ROI) to estimate the task-related FC between this ROI and the remaining ROI (267 regions) under a specific task condition. By performing this analysis across each ROI in the Shen atlas, we generated a 268 × 268 gPPI matrix for each task condition. The matrices were then transposed and averaged with the original matrices, which yielded symmetrical matrices, which were subsequently used for CPM analysis.

Regarding the use of two separate GLMs for the gPPI analysis, our study aimed to define the task-related FC under two conditions: Go trials and Successful stop trials. The first GLM including Go trials was built to estimate the gPPI during Go trials. However, due to the high frequency of Go trials in the stop signal task, it is common to regard the Go trials as an implicit baseline, as in previous IMAGEN studies (D'Alberto et al., 2018; Whelan et al., 2012). Therefore, to achieve a more accurate estimation of FC during Successful stop trials, we built a second GLM specifically for these trials. Accordingly, we have updated it in the Method Section in the main text on Page 16.

Method

2.5 Generalized psychophysiological interaction (gPPI) analysis

In this study, we adopted gPPI analysis to generate task-related FC matrices and applied CPM analysis to investigate predictive brain networks from adolescents to young adults. PPI analysis describes task-dependent FC between brain regions, traditionally examining connectivity between a seed region of interest (ROI) and the voxels of the whole rest brain. However, this study conducted a generalized PPI analysis, which is on ROI-to-ROI basis (Di et al., 2021), to yield a gPPI matrix across the whole brain instead of just a single seed region.

Given the high frequency of Go trials in SST, it is common to treat Go trials as an implicit baseline in previous IMAGEN studies (D'Alberto et al., 2018; Whelan et al., 2012). Hence, we built a separate GLM for Successful stop trials, which included two task regressors (Failed and Successful stop trials) and 36 nuisance regressors.

(6) Why did the author use PPI to construct the network, rather than the other similar methods, for example, beta series correlation (BSC)?

Thanks for your question. PPI is an approach used to calculate the functional connectivity (FC) under a specific task (i.e., task-related FC). Although most brain connectomic research has utilized resting-state FC (e.g., beta series correlation), FC during task performance has demonstrated superiority in predicting individual behaviours and traits, due to its potential to capture more behaviourally relevant information (Dhamala et al., 2022; Greene et al., 2018; Yoo et al., 2018). Specifically, Zhao et al. (2023) suggested that task-related FC outperforms both typical task-based and resting-state FC in predicting individual differences. Therefore, we chose to use task-related FC to predict sustained attention over time. We have updated it in the Introduction on Page 5.

Introduction

Although most brain connectomic research has utilized resting-state fMRI data, functional connectivity (FC) during task performance has demonstrated superiority in predicting individual behaviours and traits, due to its potential to capture more behaviourally relevant information (Dhamala et al., 2022; Greene et al., 2018; Yoo et al., 2018). Specifically, Zhao et al. (2023) suggested that task-related FC outperforms both typical task-based and resting-state FC in predicting individual differences. Hence, we applied task-related FC to predict sustained attention over time.

(7) In the section of 'Correlation analysis between the network strength and substance use', the author just described that 'the correlations between xx and xx are shown in Fig5X', and repeated it three times for three correlation results. What exactly are the results? The author should describe the results in detail. And I am wondering whether there are scatter plots for these correlation analyses?

We’d like to clarify the results in Fig. 5. Fig. 5 illustrates the significant correlations between behaviour and brain activity associated with sustained attention and Cigarette and cannabis use (Cig+CB) after FDR correction. Panel A shows the significant correlation between behaviour level of sustained attention and Cig+CB. Panels B and C show the correlations between brain activity associated with sustained attention and Cig+CB. While Panel B presents the brain activity derived from Go trials, Panel C presents brain activity derived from Successful stop trials. In response to your suggestion, we have described these results in detail on Page 9. We also have included scatter plots for the significant correlations, which are shown in Fig. 5 in Supplementary materials (Fig. S10).

Results

(6) Correlation between behaviour and brain to cannabis and cigarette use

Figs. 5A-C summarizes the results showing the correlation between ICV/brain activity and Cig+CB per timepoint and across timepoints. Fig. 5A shows correlations between ICV and Cig+CB (Tables S14-15). ICV was correlated with Cig+CB at ages 19 (Rho = 0.13, P < 0.001) and 23 (Rho = 0.17, P < 0.001). ICV at ages 14 (Rho = 0.13, P = 0.007) and 19 (Rho = 0.13, P = 0.0003) were correlated with Cig+CB at age 23. Cig+CB at age 19 was correlated with ICV at age 23 (Rho = 0.13, P = 9.38E-05). Fig. 5B shows correlations between brain activity derived from Go trials and Cig+CB (Tables S18-19). Brain activities of positive and negative networks derived from Go trials were correlated with Cig+CB at age 23 (positive network: Rhop = 0.12, P < 0.001; negative network: Rhon = -0.11, P < 0.001). Brain activity of the negative network derived from Go trials at age 14 was correlated with Cig+CB at age 23 (Rhon = -0.16, P = 0.001). Cig+CB at age 19 was correlated with brain activity of the positive network derived from Go trials at age 23 (Rhop = 0.10, P = 0.002). Fig. 5C shows the correlations between brain activity derived from Successful stop and Cig+CB (Tables S18-19). Brain activities of positive and negative networks derived from Successful stop were correlated with Cig+CB at ages 19 (positive network: Rhop = 0.10, P = 0.001; negative network: Rhon = -0.08, P = 0.013) and 23 (positive network: Rhop = 0.13, P < 0.001; negative network: Rhon = -0.11, P = 0.001).

(8) Lastly, the labels of (A), (B) ... in the figure captions are unclear. The authors should find a better way to place the labels in the caption and keep them consistent throughout all figures.

Thank you for this valuable comment. We have revised the figure captions in the main text to ensure the labels (A), (B), etc., are placed more clearly and consistently across all figures.

**Reviewer #2 (Public Review):**
While the study largely achieves its aims, several points merit further clarification:(1) Regarding connectome-based predictive modeling, an assumption is that connections associated with sustained attention remain consistent across age groups. However, this assumption might be challenged by observed differences in the sustained attention network profile (i.e., connections and related connection strength) across age groups (Figures 2 G-I, Fig. 3 G_I). It's unclear how such differences might impact the prediction results.

Thank you for your insightful comment. We’d like to clarify that we did not assume that connections associated with sustained attention remain completely consistent across age groups. Indeed, we expected that connections would change across age groups, due to the developmental changes in brain function and structure from adolescence to adulthood. Our focus was on the consistency of individual differences in sustained attention networks over time, recognising that the actual connections within those networks may change. However, we did show that there is some consistency in the specific connections associated with sustained attention over time. Notably, this consistency markedly increases when comparing ages 19 and 23, when developmental factors are less relevant. We support our reasoning above with the following analyses:

(1) Supplementary materials (Pages 2 and 5), relevant sections highlighted here for emphasis.

Method

Comparison of predictive networks identified at one timepoint versus another

Steiger’s Z value was employed to compare predictive performances of networks identified at different timepoints. This analysis involved comparing the R values derived from networks defined at distinct ages to predict ICV at the same age. For example, we compared the r values of brain networks defined at age 14 when predicting ICV at 19 (i.e., positive network: r = 0.25, negative network: r = 0.25, combined network: r = 0.28) with those R values of brain networks defined at age 19 itself (i.e., positive network: r = 0.16, negative network: r = 0.14, combined network: r = 0.16) derived from Go trials using Steiger's Z test (age 14 → age 19 vs. age 19 → 19). Similarly, comparisons were made between networks defined at age 14 predicting ICV at age 23 and those at age 23 predicting ICV at age 23 (age 14 → age 23 vs. age 23 → 23), as well as between networks defined at age 19 predicting ICV at age 23 and those at age 23 predicting ICV at age 23 (age 19 -> age 23 vs. age 23 -> age 23). These comparisons were performed separately for Go trials and Successful Stop trials.

Results

Comparison of predictive performance at different timepoints

For positive, negative, and combined networks predicting ICV derived from Go trials at age 19, the R values were higher when using predictive networks defined at 19 than those defined at 14 (Z = 3.79, Z = 3.39, Z = 3.99, all P < 0.00071). Similarly, the R values for positive, negative, and combined networks predicting ICV derived from Go trials at age 23 were higher when using predictive networks defined at age 23 compared to those defined at ages 14 (Z = 6.00, Z = 5.96, Z = 6.67, all P < 3.47e-9) or 19 (Z = 2.80, Z = 2.36, Z = 2.57, all P < 0.005).

At age 19, the R value for the positive network predicting ICV derived from Successful stop trials was higher when using predictive networks defined at 19 compared to those defined at 14 (Z = 1.54, P = 0.022), while the negative and combined networks did not show a significant difference (Z = 0.85, P = 0.398; Z = 2.29, P = 0.123). At age 23, R values for the positive and combined networks predicting ICV derived from Successful stop trials were higher when using predictive networks defined at 23 compared to those defined at 14 (Z = 3.00, Z = 2.48, all P < 3.47e-9) or 19 (Z = 2.52, Z = 1.99, all P < 0.005). However, the R value for the negative network at age 23 did not significantly differ when using predictive networks defined at 14 (Z = 1.80, P = 0.072) or 19 (Z = 1.48, P = 0.138).

These results indicate that some specific pairwise connections associated with sustained attention at earlier ages, such as 14 and 19, are still relevant as individuals grow older. However, some connections are not optimal for good sustained attention at older ages. That is, the brain reorganizes its connection patterns to maintain optimal functionality for sustained attention as it matures.

(2) Consistency of Individual Differences:

We found individual differences in ICV were significantly correlated between the three timepoints (Fig. 1B). In addition, we calculated the correlations of network strength of predictive networks predicting sustained attention derived from Go trials and Successful trials between each timepoints. We found that the correlations of network strength for predictive networks (derived from Go trials and Successful trials) were also significant (all P < 0.003). We have updated these results in the main text (Pages 7-8) and Supplementary Materials (Table S7).

(2) Cross-sectional brain connectivity

In addition, we found that network strength of positive, negative, and combined networks derived from Go trials was significantly correlated between the three timepoints (Table S7, all P < 0.003).

In addition, we found that network strength of positive, negative, and combined networks derived from Successful stop trials was significantly correlated between the three timepoints (Table S7, all P < 0.001).

(3) Predictive networks across timepoints: Predictive networks defined at age 14 were successfully applied to predict ICV at ages 19 and 23. Similarly, predictive networks defined at age 19 were successfully applied to predict ICV at age 23 (Fig. 4). These results reflect the robustness of the brain network associated with sustained attention over time.

(4) Dice coefficient analysis: We calculated the Dice coefficient to quantify the similarity of predictive networks across the three timepoints. Connections in the sustained attention networks were significantly similar from ages 14 to 23 (Table S13), despite relatively few overlapping edges over time (as discussed in Supplementary Materials on Page 6).

(5) Global brain activation: Based on these findings, we indicate that sustained attention relies on global brain activation (i.e., network strength) rather than specific regions or networks (see also (Zhao et al., 2021)).

In summary, brain network connections undergo change and are not completely consistent across time. However, individual differences in sustained attention and its network are consistent across time, as we found that (1) the brain reorganizes its connection patterns to maintain optimal functionality for sustained attention as it matures. (2) ICV and network strength of sustained attention network were significantly correlated between each timepoint. (3) Sustained attention networks identified from previous timepoints could predict ICV in the subsequent timepoint. (4) Dice coefficient analysis indicated that the edges in the sustained attention networks were significantly similar from ages 14 to 23. (5) Sustained attention networks function as a global activation, rather than specific regions or networks.

(2) Another assumption of the connectome-based predictive modeling is that the relationship between sustained attention network and substance use is linear and remains linear over development. Such linear evidence from either the literature or their data would be of help.

Thanks for your valuable suggestion. We'd like to clarify that while CPM assumes a linear relationship between brain and behaviour (Shen et al., 2017), it does not assume that the relationship between the sustained attention network and substance use remains linear over development.

Our approach in applying CPM to predict sustained attention across different timepoints was based on previous neuroimaging studies (Rosenberg et al., 2016; Rosenberg et al., 2020), which indicated linear associations between brain connectivity patterns and sustained attention using CPM analysis. These findings support the notion of a linear relationship between brain connectivity and sustained attention. In this study, we performed CPM analysis to identify predictive networks predicting sustained attention, not substance use and used the network strength of these predictive networks to represent sustained attention activity.

To examine the relationship between substance use and sustained attention, as well as its associated brain activity, we conducted correlation analyses and utilized a latent change score model instead of CPM analysis. This decision was informed by cross-sectional studies (Broyd et al., 2016; Lisdahl and Price, 2012) that consistently reported linear associations between substance use and impairments in sustained attention. Additionally, longitudinal research by (Harakeh et al., 2012) indicated a linear relationship between poorer sustained attention and the initiation and escalation of substance use over time.

Given these previous findings, we assumed a linear relationship between sustained attention and substance use. Our analyses included calculating correlations between substance use and sustained attention, as well as its associated brain activity at each timepoint and across timepoints (Fig. 5). Furthermore, we employed a three-wave bivariable latent change score model, a longitudinal approach, to assess the relationship between substance use and behavirour and brain activity associated with sustained attention (Figs. 6-7). We have added more information in the Introduction to make it more clear on Page 6.

Introduction

Additionally, previous cross-sectional and longitudinal studies (Broyd et al., 2016; Harakeh et al., 2012; Lisdahl and Price, 2012) have shown that there are linear relationships between substance use and sustained attention over time. We therefore employed correlation analyses and a latent change score model to estimate the relationship between substance use and both behaviours and brain activity associated with sustained attention.

(3) Heterogeneity in results suggests individual variability that is not fully captured by group-level analyses. For instance, Figure 1A shows decreasing ICV (better-sustained attention) with age on the group level, while there are both increasing and decreasing patterns on the individual level via visual inspection. Figure 7 demonstrates another example in which the group with a high level of sustained attention has a lower risk of substance use at a later age compared to that in the group with a low level of sustained attention. However, there are individuals in the high sustained attention group who have substance use scores as high as those in the low sustained attention group. This is important to take into consideration and could be a potential future direction for research.

Thanks for this valuable comment. We appreciate your observation regarding the individual variability that is not fully captured by group-level analyses to some degree. Fig. 1A shows the results from a linear mixed model, which explains group-level changes over time while accounting for the random effect within subjects. Similarly, Fig. 7 shows the group-level association between substance use and sustained attention. We agree that future research could indeed consider individual variability. For example, participants could be categorized based on their consistent trajectories of ICV or substance use (i.e., keep decreasing/increasing) over multiple timepoints. We agree that incorporating individual-level analyses in the future could provide valuable insights and are grateful for your suggestion, which will inform our future research directions.

The above-mentioned points might partly explain the significant but low correlations between the observed and predicted ICV as shown in Figure 4. Addressing these limitations would help enhance the study's conclusions and guide future research efforts.

We have updated the text in the Discussion on Page 13:

Discussion

However, there are still some individual variabilities not captured in this study, which could be attributed to the diversity in genetic, environmental, and developmental factors influencing sustained attention and substance use. Future research should aim to explore these variabilities in greater depth to gain better understanding of the relationship between sustained attention and substance use.

**Reviewer #3 (Public Review):**
Weaknesses: It's questionable whether the prediction approach (i.e., CPM), even when combined with longitudinal data, can establish causality. I recommend removing the term 'consequence' in the abstract and replacing it with 'predict'. Additionally, the paper could benefit from enhanced rigor through additional analyses, such as testing various thresholds and conducting lagged effect analyses with covariate regression.

Thank you for your comment. We have replaced “consequence” by “predict” in the abstract.

Abstract

Previous studies were predominantly cross-sectional or under-powered and could not indicate if impairment in sustained attention was a predictor of substance-use or a marker of the inclination to engage in such behaviour.

**Reviewer #3 (Recommendations For The Authors):**
(1) The connectivity analysis predicts both baseline and longitudinal attention measures. However, given the high correlation in attention abilities across the three time-points, it's unclear whether the connectivity predicts shared variations of attention across three time points. It would be insightful to assess if predictions at the 2nd and 3rd-time points remained significant after controlling for attention abilities at the initial time point.

Thanks for your comments. We performed the CPM analysis to predict ICV at the 2nd and 3rd timepoint, controlling for ICV at age 14 as a covariate. We found that controlling for ICV at age 14, positive, negative, and combined networks derived from Successful stop trials defined at age 14 still predicted ICV at ages 19 and 23. In addition, positive, negative, and combined networks derived from Successful stop trials defined at age 19 predicted ICV at age 23. In addition, positive, negative, and combined networks derived from Go trials defined at age 19 still predicted ICV at age 23, after controlling for ICV at age 14. However, positive, negative, and combined networks derived from Go trials defined at age 14 had lower predictive performances in predicting ICV at ages 19 and 23, after controlling for ICV at age 14. Notably, controlling for ICV at the initial timepoint did not significantly impact the performances of predictive networks derived from Successful stop trials. Accordingly, we have added this analysis and the results in the Supplementary Materials (Pages 3 and 5).

Method

Prediction across timepoints controlling for ICV at age 14

To examine whether connectivity predictors shared variations of sustained attention across timepoints, we applied predictive models developed at ages 14 and 19 to predict ICV at subsequent timepoints controlling for ICV at age 14. Specifically, we used predictive models (including parameters and selected edges) developed at age 14 to predict ICV at ages 19 and 23 separately. First, we calculated the network strength using the gPPI matrix at ages 19 and 23 based on the selected edges identified from CPM analysis at age 14. We then estimated the predicted ICV at ages 19 and 23 by applying the linear model parameters (slope and intercept) obtained from CPM analysis at age 14 to the network strength. Finally, we evaluated the predictive performance by calculating the partial correlation between the predicted and observed values at ages 19 and 23, controlling for ICV at age 14. Similarly, we applied models developed at age 19 to predict ICV at age 23, also controlling for ICV at age 14. To assess the significance of the predictive performance, we used a permutation test, shuffling the predicted ICV values and calculating partial correlation to general a random distribution over 1,000 iterations.

Results

Predictions across timepoints controlling for ICV at age 14

Positive and combined networks derived from Go trials defined at age 14 predicted ICV at ages 19 (r = 0.10, P = 0.028; r = 0.08, P = 0.047) but negative network did not (r = 0.06, P = 0.119). Positive network derived from Go trials defined at age 14 predicted ICV at age 23 (r = 0.11, P = 0.013) but negative and combined networks did not (r = 0.04, P = 0.187; r = 0.08, P = 0.056). Positive, negative, and combined networks derived from Go trials defined at age 19 predicted ICV at age 23 (r = 0.22, r = 0.19, and r = 0.22, respectively, all P < 0.001).

Positive, negative, and combined networks derived from Successful stop trials defined at age 14 predicted ICV at age 19 (r = 0.08, P = 0.036; r = 0.10, P = 0.012; r = 0.11, P = 0.009) and 23 (r = 0.11, P = 0.005; r = 0.13, P = 0.005; r = 0.13, P = 0.017) respectively. Positive, negative, and combined networks derived from Successful stop trials defined at age 19 predicted ICV at age 23 (r = 0.18, r = 0.18, and r = 0.17, respectively, all P < 0.001).

(2) In the Results section, a significance threshold of p = 0.01 was used for the CPM analysis. It would be beneficial to test the stability of these findings using alternative thresholds such as p = 0.05 or p = 0.005.

We appreciate this insightful comment. We appreciate the suggestion to test the stability of our findings using alternative significance thresholds. Indeed, we have already conducted CPM analyses using a range of thresholds, including 0.1, 0.05, 0.01, 0.005, 0.001, 0.0005, and 0.0001 (see Table S8 in supplementary Materials). The results were similar across different thresholds. Following prior studies (Feng et al., 2024; Ren et al., 2021; Yoo et al., 2018) which used P < 0.01 for feature selection, we chose to focus on the threshold of P < 0.01 for our main analysis. Following your suggestion, we have highlighted this in the Method section on Pages 17-18.

Method

2.6.1 ICV prediction

The r value with an associated P value for each edge was obtained, and a threshold P = 0.01 (Feng et al., 2024; Ren et al., 2021; Yoo et al., 2018) was set to select edges.

2.6.2 Three cross-validation schemes

In addition, we conducted the CPM analysis using a range of thresholds for feature selection and observed similar results across different thresholds (See Supplementary Materials Table S8).

(3) Could you clarify if you used one sub-sample to extract connectivity related to sustained attention and then used another sub-sample to predict substance use with attention-related connectivity?

Thank you very much for the question. We used the same sample to extract the brain network strength and estimated the correlation with substance use using both the Spearman correlation and latent change score model across three timepoints. We controlled for covariates including sex, age, and scan site at the same time. Accordingly, we have clarified this in the Method section on Page 20. We note that the CPM analyses were conducted using cross-validation, plus a leave-site-out analysis.

Method

2.7.3 Correlation between network strength and substance use

It is worth noting that all the correlations between substance use and sustained attention were conducted using the same sample across three timepoints.

(4) Could you clarify whether you have regressed covariates in the lagged effects analysis of part 7?

Thanks for this question. Yes, we confirmed that we controlled the covariates including age, sex and scan sites in the latent change score model. We have described them more clearly now in the Method section (Page 18).

Method

2.7.3 Correlation between network strength and substance use

Additionally, cross-lagged dynamic coupling (i.e., bidirectionality) was employed to explore individual differences in the relationships between substance use and linear changes in ICV/brain activity, as well as the relationship between ICV/brain activity and linear change in substance use. The model accounted for covariates such as age, sex and scan sites.

References:

Broyd, S.J., van Hell, H.H., Beale, C., Yucel, M., Solowij, N., 2016. Acute and Chronic Effects of Cannabinoids on Human Cognition-A Systematic Review. Biol Psychiatry 79, 557-567.

Chamberlain, S.R., Odlaug, B.L., Schreiber, L.R.N., Grant, J.E., 2012. Association between Tobacco Smoking and Cognitive Functioning in Young Adults. The American Journal on Addictions 21, S14-S19.

Crean, R.D., Crane, N.A., Mason, B.J., 2011. An evidence based review of acute and long-term effects of cannabis use on executive cognitive functions. J Addict Med 5, 1-8.

D'Alberto, N., Chaarani, B., Orr, C.A., Spechler, P.A., Albaugh, M.D., Allgaier, N., Wonnell, A., Banaschewski, T., Bokde, A.L.W., Bromberg, U., Buchel, C., Quinlan, E.B., Conrod, P.J., Desrivieres, S., Flor, H., Frohner, J.H., Frouin, V., Gowland, P., Heinz, A., Itterman, B., Martinot, J.L., Paillere Martinot, M.L., Artiges, E., Nees, F., Papadopoulos Orfanos, D., Poustka, L., Robbins, T.W., Smolka, M.N., Walter, H., Whelan, R., Schumann, G., Potter, A.S., Garavan, H., 2018. Individual differences in stop-related activity are inflated by the adaptive algorithm in the stop signal task. Hum Brain Mapp 39, 3263-3276.

Dhamala, E., Yeo, B.T.T., Holmes, A.J., 2022. Methodological Considerations for Brain-Based Predictive Modelling in Psychiatry. Biological Psychiatry.

Di, X., Zhang, Z.G., Biswal, B.B., 2021. Understanding psychophysiological interaction and its relations to beta series correlation. Brain Imaging and Behavior 15, 958-973.

Dougherty, D.M., Mathias, C.W., Dawes, M.A., Furr, R.M., Charles, N.E., Liguori, A., Shannon, E.E., Acheson, A., 2013. Impulsivity, attention, memory, and decision-making among adolescent marijuana users. Psychopharmacology (Berl) 226, 307-319.

Esterman, M., Rothlein, D., 2019. Models of sustained attention. Curr Opin Psychol 29, 174-180.

Feng, Q., Ren, Z., Wei, D., Liu, C., Wang, X., Li, X., Tie, B., Tang, S., Qiu, J., 2024. Connectome-based predictive modeling of Internet addiction symptomatology. Soc Cogn Affect Neurosci 19.

Greene, A.S., Gao, S., Scheinost, D., Constable, R.T., 2018. Task-induced brain state manipulation improves prediction of individual traits. Nature Communications 9, 2807.

Harakeh, Z., de Sonneville, L., van den Eijnden, R.J., Huizink, A.C., Reijneveld, S.A., Ormel, J., Verhulst, F.C., Monshouwer, K., Vollebergh, W.A., 2012. The association between neurocognitive functioning and smoking in adolescence: the TRAILS study. Neuropsychology 26, 541-550.

Hart, C.L., van Gorp, W., Haney, M., Foltin, R.W., Fischman, M.W., 2001. = . Neuropsychopharmacology 25, 757-765.

Lawrence, N.S., Ross, T.J., Stein, E.A., 2002. Cognitive mechanisms of nicotine on visual attention. Neuron 36, 539-548.

Lisdahl, K.M., Price, J.S., 2012. Increased marijuana use and gender predict poorer cognitive functioning in adolescents and emerging adults. J Int Neuropsychol Soc 18, 678-688.

O'Halloran, L., Cao, Z.P., Ruddy, K., Jollans, L., Albaugh, M.D., Aleni, A., Potter, A.S., Vahey, N., Banaschewski, T., Hohmann, S., Bokde, A.L.W., Bromberg, U., Buchel, C., Quinlan, E.B., Desrivieres, S., Flor, H., Frouin, V., Gowland, P., Heinz, A., Ittermann, B., Nees, F., Orfanos, D.P., Paus, T., Smolka, M.N., Walter, H., Schumann, G., Garavan, H., Kelly, C., Whelan, R., 2018. Neural circuitry underlying sustained attention in healthy adolescents and in ADHD symptomatology. Neuroimage 169, 395-406.

Potter, A.S., Newhouse, P.A., 2008. Acute nicotine improves cognitive deficits in young adults with attention-deficit/hyperactivity disorder. Pharmacol Biochem Behav 88, 407-417.

Ren, Z., Daker, R.J., Shi, L., Sun, J., Beaty, R.E., Wu, X., Chen, Q., Yang, W., Lyons, I.M., Green, A.E., Qiu, J., 2021. Connectome-Based Predictive Modeling of Creativity Anxiety. Neuroimage 225, 117469.

Rosenberg, M.D., Finn, E.S., Scheinost, D., Papademetris, X., Shen, X., Constable, R.T., Chun, M.M., 2016. A neuromarker of sustained attention from whole-brain functional connectivity. Nat Neurosci 19, 165-171.

Rosenberg, M.D., Scheinost, D., Greene, A.S., Avery, E.W., Kwon, Y.H., Finn, E.S., Ramani, R., Qiu, M., Constable, R.T., Chun, M.M., 2020. Functional connectivity predicts changes in attention observed across minutes, days, and months. Proc Natl Acad Sci U S A 117, 3797-3807.

Shen, X., Finn, E.S., Scheinost, D., Rosenberg, M.D., Chun, M.M., Papademetris, X., Constable, R.T., 2017. Using connectome-based predictive modeling to predict individual behavior from brain connectivity. Nat Protoc 12, 506-518.

Valentine, G., Sofuoglu, M., 2018. Cognitive Effects of Nicotine: Recent Progress. Curr Neuropharmacol 16, 403-414.

Verbruggen, F., Aron, A.R., Band, G.P.H., Beste, C., Bissett, P.G., Brockett, A.T., Brown, J.W., Chamberlain, S.R., Chambers, C.D., Colonius, H., Colzato, L.S., Corneil, B.D., Coxon, J.P., Dupuis, A., Eagle, D.M., Garavan, H., Greenhouse, I., Heathcote, A., Huster, R.J., Jahfari, S., Kenemans, J.L., Leunissen, I., Li, C.S.R., Logan, G.D., Matzke, D., Morein-Zamir, S., Murthy, A., Pare, M., Poldrack, R.A., Ridderinkhof, K.R., Robbins, T.W., Roesch, M.R., Rubia, K., Schachar, R.J., Schall, J.D., Stock, A.K., Swann, N.C., Thakkar, K.N., van der Molen, M.W., Vermeylen, L., Vink, M., Wessel, J.R., Whelan, R., Zandbelt, B.B., Boehler, C.N., 2019. A consensus guide to capturing the ability to inhibit actions and impulsive behaviors in the stop-signal task. Elife 8.

Whelan, R., Conrod, P.J., Poline, J.B., Lourdusamy, A., Banaschewski, T., Barker, G.J., Bellgrove, M.A., Buchel, C., Byrne, M., Cummins, T.D., Fauth-Buhler, M., Flor, H., Gallinat, J., Heinz, A., Ittermann, B., Mann, K., Martinot, J.L., Lalor, E.C., Lathrop, M., Loth, E., Nees, F., Paus, T., Rietschel, M., Smolka, M.N., Spanagel, R., Stephens, D.N., Struve, M., Thyreau, B., Vollstaedt-Klein, S., Robbins, T.W., Schumann, G., Garavan, H., Consortium, I., 2012. Adolescent impulsivity phenotypes characterized by distinct brain networks. Nat Neurosci 15, 920-925.

Yoo, K., Rosenberg, M.D., Hsu, W.T., Zhang, S., Li, C.R., Scheinost, D., Constable, R.T., Chun, M.M., 2018. Connectome-based predictive modeling of attention: Comparing different functional connectivity features and prediction methods across datasets. Neuroimage 167, 11-22.

Young, J.W., Finlayson, K., Spratt, C., Marston, H.M., Crawford, N., Kelly, J.S., Sharkey, J., 2004. Nicotine improves sustained attention in mice: evidence for involvement of the alpha7 nicotinic acetylcholine receptor. Neuropsychopharmacology 29, 891-900.

Zhao, W., Makowski, C., Hagler, D.J., Garavan, H.P., Thompson, W.K., Greene, D.J., Jernigan, T.L., Dale, A.M., 2023. Task fMRI paradigms may capture more behaviorally relevant information than resting-state functional connectivity. Neuroimage, 119946.

Zhao, W., Palmer, C.E., Thompson, W.K., Chaarani, B., Garavan, H.P., Casey, B.J., Jernigan, T.L., Dale, A.M., Fan, C.C., 2021. Individual Differences in Cognitive Performance Are Better Predicted by Global Rather Than Localized BOLD Activity Patterns Across the Cortex. Cereb Cortex 31, 1478-1488.